# A simple thermodynamic description of phase separation of Nup98 FG domains

Sheung Chun Ng [1] & Dirk Görlich [1] ✉

The permeability barrier of nuclear pore complexes (NPCs) controls nucleo-cytoplasmic transport. It retains inert macromolecules but allows facilitated passage of nuclear transport receptors that shuttle cargoes into or out of nuclei. The barrier can be described as a condensed phase assembled from cohesive FG repeat domains, including foremost the charge-depleted FG domain of Nup98. We found that Nup98 FG domains show an LCST-type phase separation, and we provide comprehensive and orthogonal experimental datasets for a quantitative description of this behaviour. A derived thermodynamic model correlates saturation concentration with repeat number, temperature, and ionic strength. It allows estimating the enthalpy, entropy, and $\Delta G$ (0.2 kJ/mol, 0.1 $k_B \cdot T$) contributions per repeat to phase separation and inter-repeat cohesion. While changing the cohesion strength strongly impacts the strictness of barrier, these numbers provide boundary conditions for in-depth modelling not only of barrier assembly but also of NPC passage.

Nuclear pore complexes (NPCs) are embedded into the nuclear envel-ope (NE) and cooperate with nuclear transport receptors (NTRs) for a controlled flow of material between the nucleus and cytoplasm[1]. Their permeability barrier suppresses an intermixing of nuclear and cyto-plasmic contents but grants NTRs a rapid passage by facilitated trans-location. This allows NTRs to transfer cargo across NPCs. NTRs of the importin β-superfamily can transport cargo against concentration gra-dients. For this, they draw energy from the nucleocytoplasmic RanGTP-gradient, which favours cargo-loading on one side of the NE, discharges cargo at its destination, and thus allows NTRs to return "alone" to the starting compartment for the next round of transport. Such transport cycle transfers one Ran molecule from the nucleus to the cytoplasm. Nuclear transport factor 2 (NTF2) returns Ran to the nucleus and thus has to pass NPCs more frequently than any other NTR.

The permeability barrier sorts mobile species according to size and surface features, such that passage of "inert" material larger than ~5 nm in diameter is already severely restricted[2,3]. NTRs can, however, transport much larger cargoes, such as 60 S ribosomal subunits with 25 nm diameter[4,5]. NPCs have a tremendous transport capacity, allowing up to 1000 facilitated translocation events or a mass flow of 100 MDa per NPC per second[6]. Transit times are accordingly very short, in the range of 10 ms for small cargoes[7,8] and between 24 and 90 ms for large (pre)-ribosomal subunits[9,10].

FG repeat domains are part of FG-nucleoporins and grafted at high density at the inner ring of the NPCs[11–13]. FG repeat domains contain numerous Phe-Gly (FG) motifs[14–18]. They are intrinsically disordered[19–21] and crucial for transport selectivity: Not only do they bind NTRs during facilitated translocation[22,23], they can also engage in cohesive, multivalent interactions with each other[24,25], forming a selective FG phase[6] that is a good "solvent" for NTRs or other macro-molecules with FG affinity but excludes macromolecules that are inert towards FG repeats[3,26–28]. According to the selective phase model[6], NPC passage of mobile species includes their partitioning into the FG phase, diffusion to the *trans* side, and finally phase-exit. Indeed, NTRs greatly enhance (in a RanGTPase-controlled fashion) the solubility of their cargoes in the FG phase and thereby promote directional NPC-passage[26,27,29].

Vertebrate and yeast NPCs contain each about 10 different FG domains, which differ in their prevalent FG motifs (e.g., GLFG, FSFG, or SLFG), FG motif density, composition of the inter-FG spacers as well strength of cohesive interaction[24,25,30–32]. FG domains from Nup98 or its homologues[33] are the most cohesive[28,30] and crucial ones for main-taining the permeability barrier of NPCs[34]. They are dominated by GLFG motifs and comprise 500–700 residues. Of all FG domains, they feature the highest number (around 50) and density of FG or FG-like motifs (roughly one per 12 residues). They are extremely depleted of

[1]Department of Cellular Logistics, Max Planck Institute for Multidisciplinary Sciences, Göttingen, Germany. ✉e-mail: goerlich@mpinat.mpg.de

charged residues, experience water as a poor solvent and readily phase-separate from low μM or even sub-μM concentrations to form condensed FG phases with 200–300 mg/ml or 100–300 mM FG repeat units[28]. Such self-assembled Nup98 FG phases recapitulate the transport selectivity of NPCs very well[3,28,29]. They exclude inert macromolecules (such as mCherry) to partition coefficients below 0.1 while allowing rapid entry of NTRs and NTR·cargo complexes. NTF2, for example, reaches a partition coefficient of 2000.

As an ultimate simplification, we recently engineered an FG domain (prf.GLFG$_{52 \times 12}$) composed of a 52 times (perfectly) repeated 12mer GLFG peptide[29]. It was designed to match the conserved features of native Nup98 FG domains as closely as possible, including the high FG motif number/density, compositional bias, and charge-depletion. The FG phase assembled from prf.GLFG$_{52 \times 12}$ captures the biophysical properties and transport selectivity of the original Nup98 FG phase very well indeed[29,35] and even recapitulates RanGTPase-controlled cargo import and export scenarios[29]. It thus represents the simplest possible experimental model of NPC-typical transport selectivity.

Earlier NPC reconstitution experiments[34] showed that cohesive FG domains cannot be functionally replaced by non-cohesive variants, indicating that cohesive FG interactions are fundamentally required for barrier formation. Studies based on other experimental systems, including surface-grafted FG repeats[25,31,36–38], artificial nanopore mimicries[39–41] and atomic force microscopy on real NPCs[42] also showed the presence and essence of cohesive interactions. Moreover, there have been several attempts to describe the cohesive interactions based on classical polymer physics theories[37,38,43].

Apart from nucleoporin FG domains, many other intrinsically disordered proteins (IDPs) were found to phase separate[44–48]. The resulting biomolecular condensates, sometimes in the form of membrane-less organelles, are implicated in a wide range of biological functions and pathologies[49,50]. In many cases, phase separation behaviours of such IDPs resemble those of artificial polymers[51–55]: phase separation occurs when the concentration of the protein exceeds the saturation concentration ($C_{sat}$). The saturation concentration also corresponds to the remaining concentration in the aqueous phase when in thermodynamic equilibrium with the condensed phase. The phase separation propensity can conveniently be defined as the inverse of the saturation concentration[46–48,55].

Saturation concentration and thus phase separation propensity depend on environmental factors. UCST (Upper critical solution temperature) behaviour implies that lower temperature favours phase separation, while LCST (Lower critical solution temperature) behaviour describes cases where higher temperature favours phase separation. UCST behaviours of some phase-separating IDPs[44,47,56,57] can be approximated by the Flory-Huggins (FH) model[58,59]. This model assumes that phase separation is entropically disfavoured and that enthalpic contributions at lower temperature allow phase separation (because the entropic T·ΔS term gets smaller). On the other hand, the LCST-type phase separation is driven by the unfavourable entropy of mixing the polymer and solvent molecules[52,53] and thus does not comply to the classical FH model. Well-documented LCST examples of phase separating IDPs include elastin-like-peptides (ELPs)[52,60–63] and UBQLN2[64].

To explain the size selectivity of NPC passage, the selective phase model[6] assumed that multivalently interacting FG repeats assemble a sieve-like FG phase, which retains particles larger than the meshes but allows smaller molecules to pass through, whereby FG motifs mediate cohesive interactions[24,31,34]. As NTRs bind FG motifs, they can disengage cohesive contacts and thus "melt" through the FG meshes. Indeed, crystallographic and computational analyses revealed multiple FG-binding sites within a given NTR[65–67]. In addition to such structurally dedicated binding sites, NPC-passage can be accelerated by exposed "FG-philic" residues (such as hydrophobics, cysteine, or arginine) on the surface of a mobile species. This way, so-called GFP$^{NTR}$ variants have been engineered that pass NPCs at similar rates as native NTRs[3]. Thermodynamically speaking, cohesive interactions pose an energetic barrier against partitioning any large mobile species into the FG phase. Favourable interactions of the mobile species with FG repeats can compensate for these energetic costs and thus facilitate NPC passage. Although this model is attractive, detailed energetic parameters have been lacking so far.

Here, we approach this problem by studying the Nup98 phase system in depth. To simplify interpretations, we used the afore-described prf.GLFG$_{52 \times 12}$ as a primary experimental system. We found that Nup98 FG domains exhibit an LCST behaviour with parallels to how non-ionic surfactants (like polyoxyethylene alkyl ethers) form micelles. We developed a thermodynamic model that correlates the saturation concentration, number of repeat units, environmental temperature, and salt concentration. It also allowed to derive enthalpy, entropy, and free energy contribution per repeat unit (0.2 kJ/mol) to phase separation and to describe the strength of cohesive interactions in quantitative energy terms. Moreover, we found that the strictness of the barrier is very sensitive to changes in cohesion strength. Thus, the numbers reported here establish the basis for further quantitative modelling not only of barrier assembly but also of nuclear pore passage.

## Results

### Partitioning of FG repeats into an FG phase

Hydrophobic FG motifs are required for inter FG repeat-cohesion[24,28,30–32,34], which in turn drives the assembly of the FG phase and thus of the NPC permeability barrier. The selective phase model suggests that NTRs translocate through the phase by transiently disengaging cohesive contacts. The strength of such cohesive interactions should then be a crucial parameter for the translocation process and the selectivity of NPCs. Since the Nup98 FG domains appears the most crucial one, it is the focus of our study.

A challenge in measuring this parameter is that cohesive interactions are rather complex and associated with several unknowns. First, many cohesive units within an FG domain need to be considered. Second, given the repeats' intrinsic disorder, cohesive interactions are probably fuzzy and heterogeneous. Finally, the valency with which elementary cohesive units interact is unknown; and possibly, this valency is not even fixed but flexible. Thus, standard biochemical dissociation constants appear to be a rather inadequate description.

Yet, it seems feasible to derive energetic parameters for cohesion from the FG repeats' partitioning from the bulk buffer into an FG phase, in particular when partitioning coefficients can be correlated to the repeat number. For the cleanest possible correlation, all repeat units should contribute the same. However, such equal contribution cannot be assumed for native Nup98 FG domains that comprise variable FG repeats with heterogeneous length, FG motifs and inter-FG spacers. We, therefore, used the prf.GLFG$_{52 \times 12}$ perfectly repeated FG domain[29], which is composed of 52 connected, identical 12mer peptides (sequence: GGLFGGNTQPAT). It was engineered and validated to capture features (length, FG density, amino acid composition, dipeptide frequencies, etc.) and properties (in particular transport selectivity) of the original MacNup98 FG domain[28,29] as closely as possible. For the sake of simplicity, we also deleted the Gle2-binding sequence (GLEBS, 44 residues) that not only binds Gle2 in a cellular context but also contributes to phase separation.

For determining the energy contribution per FG repeat, we produced prf.GLFG$_{52 \times 12}$ variants with different repeat numbers ($N = 7$, 13, 18, 26, 37, 44, 52 and 70), labelled them with a single Atto488 fluorophore, and used confocal laser scanning microscopy (CLSM) to measure their partition coefficients in a given "host" FG phase (Fig. 1a, b). As our standard host, we chose prf.GLFG$_{52 \times 12}^{[+GLEBS]}$ where

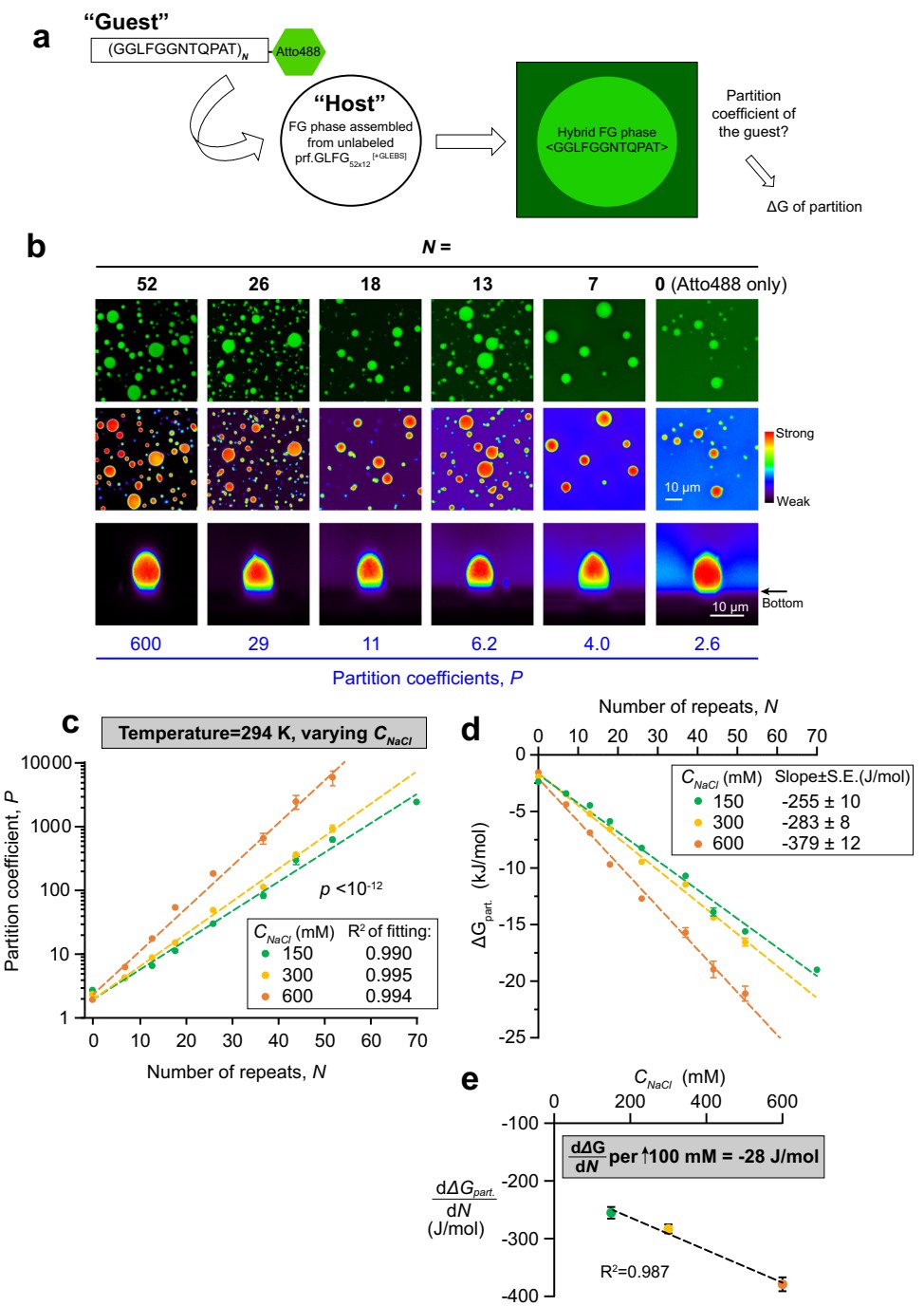

**Fig. 1 | Energy contribution per FG repeat unit in FG-FG partition. a** Scheme of the partition experiments: variants with different number ($N$) of connected perfect repeat units (as "guests") were covalently coupled with a chemical fluorophore (Atto488), and the partition coefficient, $P$ of each into a given "host" FG phase, formed by prf.GLFG$_{52\times12}$[+GLEBS], was measured by confocal laser scanning microscopy (CLSM) at 294 K. $P$ is related to Gibbs free energy of partition ($\Delta G_{part.}$) as described by Eq. 1. **b** Representative images from CLSM, showing the Atto488 signal of guests with indicated $N$ at 150 mM NaCl. Upper two panels: XY scans with green and false colour, respectively. Lower panel: XZ scans with false colour showing an FG particle in each condition. Molar ratio of host: guest and scanning settings in the measurements were adjusted individually to cover wide dynamic ranges. For comparison herein, brightness of the FG phase in the images was normalized.

**c** $P$ against $N$ was determined at different NaCl concentrations ($C_{NaCl}$) at a fixed temperature. Measurement for each condition was repeated three times with independent samples, and mean values are shown with S.D. as the error bars. Note that the logarithm of $P$ scales linearly with $N$ for all three concentrations of NaCl. A two-tailed $p$-value was computed by Analysis of Covariance (ANCOVA) to assess if the differences in slopes of the fits are statistically significant. No adjustment was made for multiple comparisons. This statistics analysis also applies to Figs. 4a, 5d, 6a and 7a. **d** $\Delta G_{part.}$ calculated from $P$ ($n=3$) by Eq. 1 was plotted against $N$ for different $C_{NaCl}$. Mean values are shown with S.D. as the error bars. Each plot can be fitted with a linear function. **e** Slopes of $\Delta G_{part.}$ against $N$ (i.e., $\frac{d\Delta G_{part.}}{dN}$) derived from (**d**) are plotted against $C_{NaCl}$. Data are presented as mean values with standard errors (S.E.) of fitting as the error bars.

the perfect repeats are still interrupted by the GLEBS domain, because this variant has lower saturation concentration of 0.3 μM[29] than prf.GLFG$_{52\times12}$ (Supp. Fig. 1), which minimizes interferences of the measurements by host molecules in the aqueous phase.

These measurements gave striking Log-linear relationships (Fig. 1c) between the partition coefficients ($P$) and the number of repeats ($N$), which is consistent with each repeat contributing equal free energy increments ($\Delta G_{repeat}$) to the partitioning (see Supp. Table 1

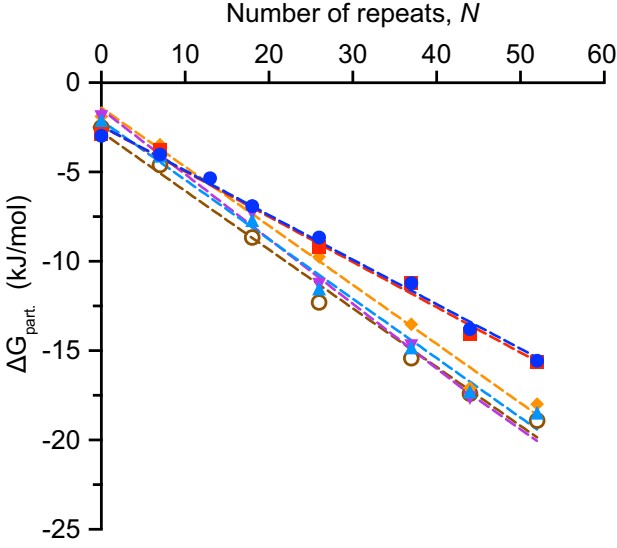

**Fig. 2 | Energy contribution per FG repeat unit in partition into FG phases assembled from various species.** Partition coefficients of the perfect repeat "guests" into different "hosts", FG phases assembled from Nup98 FG domain homologues from different eukaryotic species (*Saccharomyces cerevisiae, Sc; Tetrahymena thermophila, Tt; Xenopus tropicalis, Xt; Branchiostoma floridae, Bf; Arabidopsis thaliana, At* and *Trypanosoma brucei, Tb*), were measured at 150 mM NaCl (data see Source Data file), and $\Delta G_{part.}$ calculated by Eq. 1 was plotted against *N* for different Nup98 FG domain homologues (*n* = 3, with independent samples, mean values are shown, S.D.s are smaller than the size of the symbols). Lower panel: Slopes of the plots ($\frac{d\Delta G_{part.}}{dN}$) are presented as best-fit values ± S.E. of fitting.

for a list of symbols):

$$\Delta G_{part.} = -RT \cdot \ln P = G^0 + N \cdot \Delta G_{repeat} \qquad (1)$$

where $R$ is the gas constant (8.31 J/mol·K) and $T$ the absolute temperature in Kelvin. The $G^0$ term of the formula considers here that the fluorophore is (weakly) FG-philic[29] and compensates the bias from the labelling. The fit of the data revealed that one repeat unit contributes 255 J/mol to the partition equilibrium, when measured at 21 °C (294 K) and 150 mM NaCl. This number increased linearly with NaCl concentration (hereafter denoted as $C_{NaCl}$), to 283 J/mol at 300 mM, and to 379 J/mol at 600 mM NaCl (Fig. 1d, e). This 30 J/mol difference per 100 mM increase in $C_{NaCl}$ supports the notion that inter-FG cohesion is driven by hydrophobic interactions (which get stronger with higher salt).

Next, we repeated the experiment with the same set of guests, but in different "hosts", namely FG phases formed by wild-type Nup98 FG domains of evolutionarily distant species[28] including the yeast *Saccharomyces cerevisiae* (*Sc*Nup116[16]), the ciliate *Tetrahymena thermophila* (MacNup98A[68]; *Tt*Mac98A), the frog *Xenopus tropicalis* (*Xt*Nup98[30]), the lancelet *Branchiostoma floridae*

(*Bf*Nup98), the plant *Arabidopsis thaliana* (*At*Nup98B[69]), and the euglenozoan *Trypanosoma brucei* (*Tb*Nup158[70]) (Fig. 2). Again, we observed rather similar partition coefficients and thus $\Delta G_{part.}$ contributions per repeat unit. As the host FG phase defines the "attraction" of the guest FG repeats, this suggests that the cohesive potential of Nup98 FG domains is rather conserved in evolution.

### Log-linear scaling of FG domain saturation concentration and phase transition temperature

In the previous section, we determined the energy contribution per FG repeat by measurements of partitioning. Alternatively, the energy contribution per FG repeat can be derived from parameterization of a model of the phase separation process. To establish such a model, we started by obtaining a phase diagram that describes the phase separation conditions of prf.GLFG$_{52×12}$.

We prepared solutions of different concentrations of prf.GLFG$_{52×12}$ in 20 mM sodium phosphate (NaPi) buffers (pH maintained at 6.8) with $C_{NaCl}$ ranging from 75 to 600 mM. Already during sample preparation, we observed that phase separation of all samples was suppressed when they were put on ice. Dynamic light scattering (DLS)[61] was then applied to analyse the transition points in a temperature-controlled chamber. The onset of phase separation became evident as a sharp increase in light scattering signal–caused by the formation of μm-sized FG particles[28] (Fig. 3a), when the temperature was increased. For one, this indicates an LCST behaviour of the prf.GLFG$_{52×12}$ repeats. In addition, we measured systematically phase transition temperatures (*T*) at varying concentrations of prf.GLFG$_{52×12}$ and NaCl.

For example, at $C_{NaCl}$ = 75 mM and [prf.GLFG$_{52×12}$] = 5 μM, phase transition occurred at 33 °C (306 K, Fig. 3b). When [prf.GLFG$_{52×12}$] was increased to 20 μM, the transition temperature decreased to 22 °C (295 K). At $C_{NaCl}$ = 300 mM, the transition temperatures were lower: 25 °C (298 K) at [prf.GLFG$_{52×12}$] = 5 μM and 16 °C (289 K) at [prf.GLFG$_{52×12}$] = 20 μM (Fig. 3c). Interestingly, by plotting the transition temperature against the logarithm of prf.GLFG$_{52×12}$ concentration at given $C_{NaCl}$, we observed obvious linear relationships (R-squared values >0.96, Fig. 3d). This type of relationship was also reported for engineered ELPs[60,61].

Next, we used an alternative approach to measure the saturation concentrations (denoted as $C_{sat}$) of prf.GLFG$_{52×12}$ at different $C_{NaCl}$ (Fig. 3e): solutions of [prf.GLFG$_{52×12}$] = 20 μM were centrifuged at a given temperature, and if phase separation had happened, the condensed FG phase was pelleted and the saturation concentrations could be measured as the remaining concentrations in the supernatant (i.e., the aqueous phase). We found that the logarithm of $C_{sat}$ scales linearly with $C_{NaCl}$ at constant temperature (27 °C/300 K). Datasets shown in Fig. 3d, e are consistent with each other: for example, Fig. 3d shows a transition point at [prf.GLFG$_{52×12}$] = 5 μM, 298 K and $C_{NaCl}$ = 150 mM; and Fig. 3e documents a very similar transition point at $C_{NaCl}$ = 150 mM.

Furthermore, we found LCST behaviours not only for the prf.GLFG$_{52×12}$ repeats but also for wildtype Nup98 FG domains from animals, plants, and ciliates (Fig. 3f): the saturation concentrations decreased in all cases sharply with increasing temperatures. LCST behaviour is thus an evolutionarily conserved feature of the Nup98-related FG domains[28].

### A thermodynamic description of the FG phase behaviour

How could the above observations be rationalized/modelled? Indeed, thermodynamics have been useful to describe LCST-type phase separation behaviours[71] and physical parameters, like free energy change, could be extracted from parameterization of the thermodynamic models[72–75]. Well-established systems include micellization of non-ionic surfactants/detergents, like polyoxyethylene alkyl ethers.

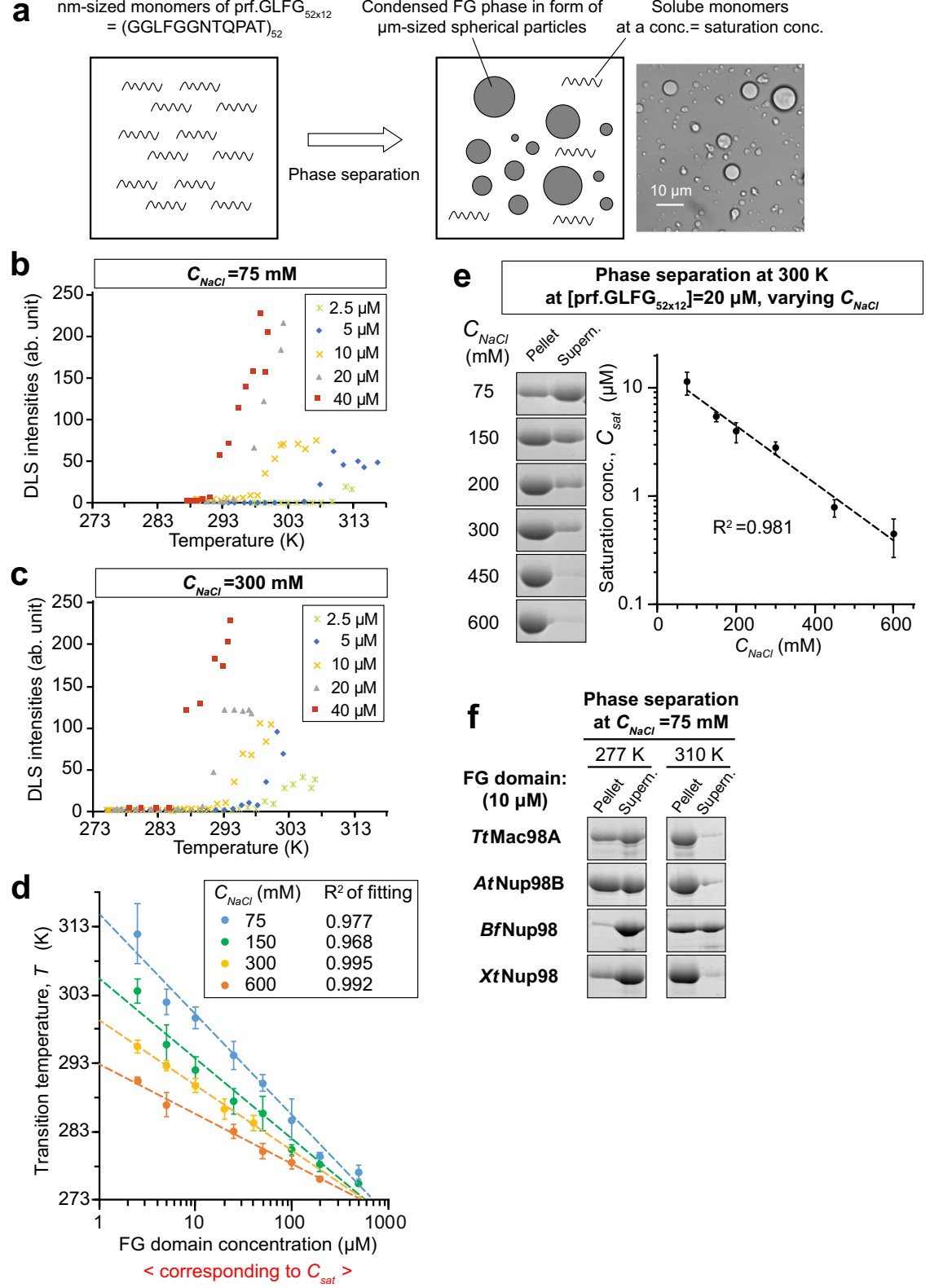

**a** nm-sized monomers of prf.GLFG$_{52x12}$ = (GGLFGGNTQPAT)$_{52}$ · Condensed FG phase in form of μm-sized spherical particles · Soluble monomers at a conc.= saturation conc.

Phase separation

10 μm

**b** $C_{NaCl}$ =75 mM

DLS intensities (ab. unit) vs Temperature (K)
- 2.5 μM
- 5 μM
- 10 μM
- 20 μM
- 40 μM

**c** $C_{NaCl}$ =300 mM

DLS intensities (ab. unit) vs Temperature (K)
- 2.5 μM
- 5 μM
- 10 μM
- 20 μM
- 40 μM

**d**

Transition temperature, $T$ (K) vs FG domain concentration (μM)

| $C_{NaCl}$ (mM) | R$^2$ of fitting |
| --- | --- |
| 75 | 0.977 |
| 150 | 0.968 |
| 300 | 0.995 |
| 600 | 0.992 |

< corresponding to $C_{sat}$ >

**e** **Phase separation at 300 K at [prf.GLFG$_{52x12}$]=20 μM, varying $C_{NaCl}$**

$C_{NaCl}$ (mM) | Pellet | Supern.
75
150
200
300
450
600

Saturation conc., $C_{sat}$ (μM) vs $C_{NaCl}$ (mM)

R$^2$ =0.981

**f** **Phase separation at $C_{NaCl}$ =75 mM**

FG domain: (10 μM)

| | 277 K | | 310 K | |
| --- | --- | --- | --- | --- |
| | Pellet | Supern. | Pellet | Supern. |
| *Tt*Mac98A | | | | |
| *At*Nup98B | | | | |
| *Bf*Nup98 | | | | |
| *Xt*Nup98 | | | | |

We applied a similar approach to describe the phase separation of prf.GLFG$_{52×12}$:

The Gibbs free energy change (ΔG) for phase separation can be derived from the distribution constant (K) of the phase equilibrium:

$$\Delta G = -RT \cdot \ln K = -RT \cdot \ln \left( \frac{C_{dense}}{C_{sat}} \right) = RT \cdot \ln \left( \frac{C_{sat}}{C_{dense}} \right) \quad (2)$$

where $C_{sat}$ is the FG domain concentration in the dilute phase and $C_{dense}$ is the concentration in the condensed phase. ΔG can also be expressed by the corresponding enthalpy (ΔH) and entropy changes (ΔS):

$$\Delta G = \Delta H - T\Delta S \quad (3)$$

**Fig. 3 | Phase separation conditions of prf.GLFG$_{52\times12}$. a** Illustration of the LCST-type phase separation in this study: At a low concentration, prf.GLFG$_{52\times12}$ (representing Nup98 FG domains) remains soluble in an aqueous solvent and a single phase was observed. When its concentration is increased to above a threshold termed as saturation concentration, $C_{sat}$, the amount of protein in the system exceeds the solvation capacity of the solvent and a synergistic assembly of protein molecules occurs spontaneously, leading to the formation of a protein-rich "FG phase", which separates from the aqueous phase. FG phase is in form of scattered micrometre-sized spherical particles ("FG particles"). The solvation capacity of the solvent and thus the saturation concentration of the FG domain decreases when the temperature increases. Therefore, phase separation may also occur when the temperature is increased to above a threshold, i.e., *transition temperature*. After phase separation has occurred, the aqueous phase still contains solvated FG domain molecules at a concentration equal to $C_{sat}$. Right: a phase-contrast microscopy image showing a typical phase-separated state. **b, c** Dilutions of prf.GLFG$_{52\times12}$ of indicated concentrations were prepared on ice in buffers containing 20 mM NaPi pH 6.8 and either 75 mM (**b**) or 300 mM NaCl (**c**). Each dilution was analysed by Dynamic Light Scattering (DLS) with increasing temperature until a sharp increase of light scattering intensity was observed, which indicates phase separation. The temperature at which phase separation had just occurred was taken as the *transition temperature*, *T*. **d** DLS analyses were repeated at indicated NaCl concentrations ($C_{NaCl}$) to determine *T*, which was plotted against [prf.GLFG$_{52\times12}$]. Measurement for each condition was repeated three times with independent samples, and mean

values are shown with S.D. as the error bars to show the variation between replicates. Each dataset for a given $C_{NaCl}$ was fitted to a simple exponential function and the best fits were shown as dashed lines. **e** 20 μM dilutions of prf.GLFG$_{52\times12}$ were prepared in buffers containing the indicated concentration of NaCl and centrifuged at the same temperature (27 °C/300 K). SDS samples of the obtained pellets (FG phase), if there were, and supernatants (soluble content) were loaded for SDS-PAGE at equal ratio (7%), followed by Coomassie blue staining for quantification. *Saturation concentrations*, $C_{sat}$, for individual conditions were equal to the concentrations of the supernatants. (For reader's convenience, note that $C_{sat}$ can be quickly estimated from the band intensities shown here: $C_{sat}$ = ratio of supernatant/ (supernatant + pellet) × assay concentration, 20 μM). $C_{sat}$ was plotted against $C_{NaCl}$. Measurement for each condition was repeated four times with independent samples, and mean values are shown with S.D. as the error bars. The mean values were fitted to a simple exponential function (dashed line). **f** 10 μM dilutions of indicated Nup98 FG domain homologues from different eukaryotic species (*Tetrahymena thermophila, Tt; Arabidopsis thaliana, At; Branchiostoma floridae, Bf and Xenopus tropicalis, Xt*) were prepared in a buffer containing 75 mM of NaCl and centrifuged at the indicated temperatures. SDS samples of the obtained pellets and supernatants were loaded for SDS-PAGE at equal ratio (7%), followed by Coomassie blue staining. The exact saturation concentrations were not determined. This experiment was repeated two times independently with similar results, and the representative gel images are shown.

Combining Eqs. 2 and 3:

$$RT \cdot \ln\left(\frac{C_{sat}}{C_{dense}}\right) = \Delta H - T\Delta S$$

$$\ln\left(\frac{C_{sat}}{C_{dense}}\right) = \frac{\Delta H}{R} \cdot \frac{1}{T} - \frac{\Delta S}{R} \tag{4}$$

We previously found that $C_{dense}$ of prf.GLFG$_{52\times12}$ is 4.5 mM or 260 mg/ml[35], which is similar to that of FG phases assembled from native Nup98 FG domains of evolutionarily diverse species[28]. As there is relatively little change of $C_{dense}$ within the tested ranges of temperature and salt concentration, we can consider $C_{dense}$ as constant (analogous to the approximation by Bremer et al.[76]).

Assuming that changes of $\Delta H$ and $\Delta S$ with temperature are negligible, a plot of ln ($C_{sat}/C_{dense}$) against $1/T$ (known as van't Hoff plot[76–85]) should give a straight line, allowing to estimate $\Delta H$ (=$slope \times R$) and $\Delta S$ (=$-intercept \times R$).

We generated van't Hoff plots (Fig. 4a) with conditions obtained from Fig. 3d, considering that at the transition temperature, the sample concentration equals to the saturation concentration ($C_{sat}$). Straight lines were observed for different $C_{NaCl}$, confirming also the validity of the assumption that $C_{dense}$, $\Delta H$, and $\Delta S$ remain sufficiently constant within the range of tested temperatures. The van't Hoff plots allowed us to derive $\Delta H$ and $\Delta S$ for different $C_{NaCl}$ (listed in Supp. Table 2). Interestingly, we found that both scaled linearly with $C_{NaCl}$ (Fig. 4b, c). Therefore,

$$\Delta H = a \cdot C_{NaCl} + b \tag{5}$$

$$\Delta S = c \cdot C_{NaCl} + d \tag{6}$$

where $a$, $b$, $c$ and $d$ are constants derived from Fig. 4b, c (Supp. Fig. 2 shows the relationships in terms of ionic strength). Combining Eqs. 3, 5 and 6 gives:

$$\Delta G = a \cdot C_{NaCl} + b - T(c \cdot C_{NaCl} + d) \tag{7}$$

or combining Eqs. 4, 5 and 6:

$$\ln\left(\frac{C_{sat}}{C_{dense}}\right) = \frac{a \cdot C_{NaCl} + b}{RT} - \frac{c \cdot C_{NaCl} + d}{R} \tag{8}$$

Now, the Gibbs free energy change or saturation concentration can be correlated to transition temperature and salt concentration by Eq. 7 or Eq. 8, respectively. A plot of Eq. 7 (Fig. 4d) or calculations of transition temperatures in different conditions (Fig. 4e) capture the experimental data very well. Even more, additional transition conditions obtained at $C_{NaCl}$ = 1200 mM can also be captured, indicating that the equation can be applied to a wide range of salt concentrations. Moreover, the model now allowed us to rationalize different datasets: e.g., the trend observed in Fig. 3e can be rationalized, because Eq. 8 can be written as:

$$\ln\left(\frac{C_{sat}}{C_{dense}}\right) = \frac{(a - T \cdot c)\,C_{NaCl} + b - T \cdot d}{RT}$$

i.e., $\ln C_{sat} \propto -C_{NaCl}$ at a fixed temperature. We can now also describe the system by energetic terms, for example, increasing the temperature from 20 to 30 °C at $C_{NaCl}$ = 150 mM leads to an increase in $\Delta G$ ($\Delta\Delta G$) by roughly 5 kJ/mol.

## Saturation concentrations decrease exponentially with increasing FG repeat numbers

We then asked how the phase separation propensity would change with the number of perfect repeats (*N*) per polypeptide. A rationale for looking into these details was to assess how increments in repeat numbers affect the phase separation behaviour.

For that, we compared the phase separation of prf.GLFG$_{52\times12}$ with some of the aforementioned variants: prf.GLFG$_{70\times12}$, prf.GLFG$_{44\times12}$, prf.GLFG$_{37\times12}$ and prf.GLFG$_{26\times12}$ (i.e., *N* = 70, 52, 44, 37 and 26) (Supp. Fig. 3). We first prepared concentrated stock solutions of each variant, and diluted each to 20 μM (polypeptide concentration) at $C_{NaCl}$ = 150 mM. Immediately we observed that the turbidity of the dilutions increased with *N* in general, while dilutions of prf.GLFG$_{37\times12}$ and prf.GLFG$_{26\times12}$ remained clear. We attempted to spin down the insoluble contents (at 27 °C/300 K) and analysed the pellets and supernatants by the approach described above. As shown in Fig. 5a, the concentrations of supernatant (i.e., = $C_{sat}$) decreased from *N* = 44 to 70. For the 37× and 26× repeats, however, no phase separation occurred

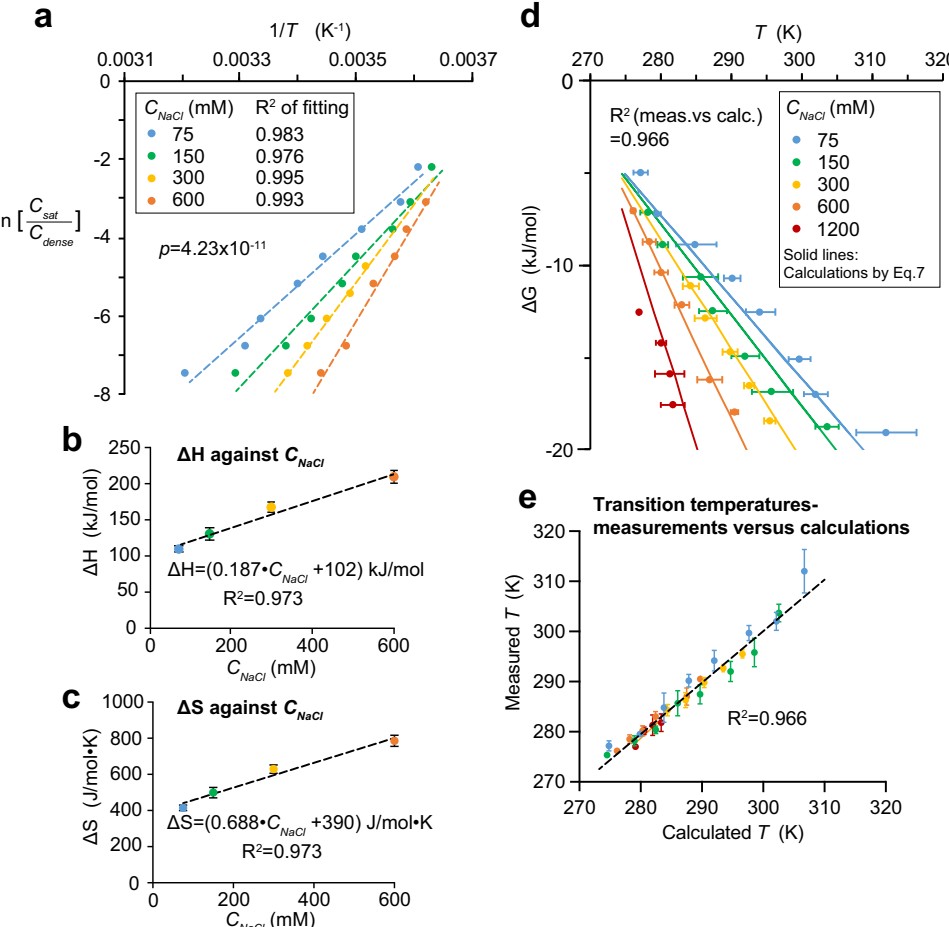

**Fig. 4 | Phase separation behaviours of prf.GLFG5₂ₓ₁₂ can be described by thermodynamics. a** ln ($C_{sat}/C_{dense}$) is plotted against 1/(*transition temperature*) for different $C_{NaCl}$ using dataset from Fig. 3d (each data point is the mean of three replicates), where *transition temperature, T* is in Kelvin. These plots are known as the van't Hoff plots (Eq. 4). Each plot can be fitted with a linear function with a high R-squared value. A two-tailed *p*-value was computed to assess if the differences in slopes of the fits are statistically significant. *ΔH* and *ΔS* for phase separation are derived from each linear fit (see Supp. Table 2, R is the gas constant). **b, c** *ΔH* and *ΔS* derived from best-fits in **a** were plotted against $C_{NaCl}$, respectively. Data are presented as mean values with S.E. of fitting as error bars (see also Supp. Table 2).

Interestingly, both *ΔH* and *ΔS* scale linearly with $C_{NaCl}$. **d** Dataset from Fig. 3d is converted to *ΔG* against *T* (in Kelvin) by Eq. 2 and a dataset measured at extra high [NaCl] (1200 mM) is added. Mean values of three replicates with independent samples are shown with S.D. as the error bars. All these data can be captured by a plot of Eq. 7 (solid lines), which is derived from a thermodynamic model and observations in (**b, c**). R-squared value here shows the degree of correlation between the dataset and calculations from Eq. 7. **e** Values of *T* at different conditions ($C_{sat}$ and $C_{NaCl}$; same colouring as in **d**) were calculated by Eq. 8 and are plotted against values obtained in experiments. R-squared value shows the degree of correlation between the calculations and experiments.

until the assay concentration was raised to 50 and 300 μM, respectively.

The ratio of $C_{dense}$ : $C_{sat}$ corresponds to a partition coefficient. And indeed, these ratios are, for a given *N*, quite similar to the partition coefficients derived from partitioning of fluorescently labelled FG repeats into a host phase (Supp. Table 3 and Fig. 1).

Moreover, in line with the previous section, phase separation of all the variants was suppressed by lowered temperatures (Fig. 5b, c). For example, 150 μM of the 26× repeats showed phase separation at 37 °C (310 K), but not at 27 °C (300 K) or lower. By quantifying $C_{sat}$ and plotting against *N* at given temperatures, we found that the logarithm of $C_{sat}$ scales linearly with *N* (Fig. 5d).

**Energy contribution to phase separation per FG repeat unit**

With the temperature dependences at different values of *N*, we constructed van't Hoff plots (Fig. 6a) as described above, assuming as before that the FG phases have approximately similar mass density (260 mg/ml) as that assembled from prf.GLFG₅₂ₓ₁₂ irrespective of the temperature. Using this approximation, *ΔH* and *ΔS* were estimated for each variant (Supp. Table 4), as introduced above. Relationships of *ΔH*

and *ΔS* with *N* can be approximated by the following linear relationships, respectively (Fig. 6b, c):

$$\Delta H = w \cdot N + x \quad (9)$$

$$\Delta S = y \cdot N + z \quad (10)$$

where *w, x, y* and *z* are constants derived from Fig. 6b, c.

Note that $w = \frac{d\Delta H}{dN}$ = the enthalpy change per repeat unit = 1.46 kJ/mol, and $y = \frac{d\Delta S}{dN}$ = the entropy change per repeat unit = 5.66 J/mol·K. With these values, we can also derive $\frac{d\Delta G}{dN}$, the Gibbs free change per repeat unit (=$\Delta G_{repeat}$) which scales linearly with the temperature (Fig. 6d, e):

$$\frac{d\Delta G}{dN} = \frac{d\Delta H}{dN} - T \cdot \frac{d\Delta S}{dN} = w - T \cdot y \quad (11)$$

For example, $\Delta G_{repeat}$ is about −200 J/mol at a temperature of 294 K. Note that this value is consistent with the energy contribution

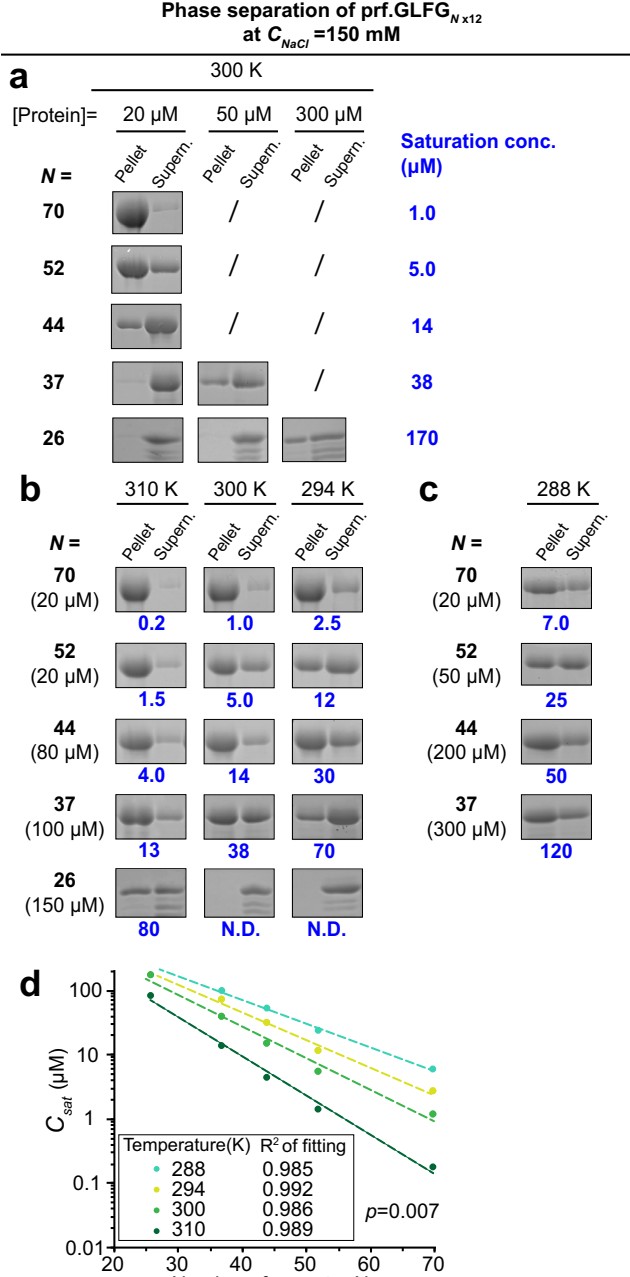

**Fig. 5 | Saturation concentrations decrease exponentially with increasing FG repeat numbers. a**–**c** Dilutions of variants containing different number of connected perfect repeats ($N$) were centrifuged and SDS samples of the obtained pellets (FG phase) and supernatants were loaded for SDS-PAGE at equal ratio. $C_{sat}$ for each condition were determined as described above, and are shown in blue. Assay temperatures and concentrations of the dilutions of the variants are as indicated. Concentration of NaCl in the assay buffer was the same, 150 mM for (**a**–**c**). Experiments for each condition were repeated two times on independent samples with similar results and the mean values are shown. Marked with "/" or "N.D": not determined. Full scans of gels with molecular weight markers are provided in the Source data file. **d** $C_{sat}$ obtained above at different temperatures were plotted against the number of perfect repeats ($N$). Each dataset for a given temperature can be fitted with a simple exponential function.

per repeat to the partition equilibrium obtained in the previous section (Fig. 1d), but derived with an orthogonal method. On the other hand, we can also correlate the Gibbs free change for phase separation and

the number of repeats by combining Eqs. 3, 9 and 10:

$$\Delta G = w \cdot N + x - T(y \cdot N + z)$$

$$\Delta G = (w - T \cdot y) \cdot N - T \cdot z + x \qquad (12)$$

These formulas capture the incremental contributions per FG repeat towards the total free energy change for phase separation. A plot of Eq. 12 captures the experimental data very well (R-squared value >0.99, Fig. 6d). Moreover, we expanded the measurements to higher NaCl concentrations ($C_{NaCl}$ = 300 and 600 mM, Fig. 7), and found that the magnitude of $\Delta G_{repeat}$ scales linearly with $C_{NaCl}$: roughly −40 J/mol difference per 100 mM increase in $C_{NaCl}$ at 300 K. This gradient is again consistent with that derived from the partition experiments (Fig. 1e), which were performed at a slightly lower temperature (294 K).

### Tuning the FG phase transport selectivity with salt concentration

Inter-FG cohesion not only allows the assembly of a dense FG phase, but it is also likely the key to transport selectivity since mobile species must interrupt cohesive interactions transiently for entering the phase. This transient interruption can be seen as an energetic penalty−compensable by favourable NTR-FG interactions.

Considering this, we tested if enhanced inter FG-cohesion, induced by higher salt concentrations, would impact the partitioning of mobile species into FG phases. With partition coefficients of ≤0.05, the inert species mCherry remained perfectly excluded at all salt concentrations (see Fig. 8a for prf.GLFG$_{52×12}$[+GLEBS] and Supp. Fig. 5 for wild-type Nup98 FG phases of *Tt*Mac98A and *Sc*Nup116). This does not rule out a stricter exclusion at higher salt, because the assay (based on confocal fluorescence microscopy) cannot discern partition coefficients lower than 0.05. There was also little effect on the very high partition coefficients (≥2000) of the small (30 kDa) nuclear transport receptor NTF2 (Fig. 8b), perhaps because salt enhances not only cohesion but concomitantly also hydrophobic FG·NTF2 interactions. Partition coefficients of larger (110−130 kDa) species with optimized FG-philic surfaces also remained very high (>1400) at elevated salt concentration (Fig. 8c, d). This applied to the GFP[NTR]_3B7C variant−a tetramer engineered for rapid NPC passage[3]−as well as to importin β carrying the very FG-philic IBB-sffrGFP7 cargo[3].

However, we observed a striking effect for a fusion of importin β to another GFP variant (shGFP2; ref. 3) (Fig. 8e and Supp. Fig. 5). In this case, the GFP had been engineered to reach the other extreme, i.e., to be highly repellent for FG phases ("FG-phobic"). The partition coefficient of this fusion dropped nearly 100-fold from 52 at 50 mM NaCl to 0.6 at 300 mM. A similar effect was seen for an IBB-EGFP fusion bound to importin β, and also when the perfect FG repeats were replaced with authentic Nup98 FG domains (Supp. Fig. 5). We interpret this drastic effect as a consequence of the salt-induced increase in inter-FG repeat cohesion, which makes, in particular, the entry of the fused FG-phobic GFP moiety energetically more costly. In any case, this demonstrates that tuning FG cohesion can have a striking impact on transport selectivity, particularly for the transport of less FG-philic species, like an NTR carrying (a large) cargo with an exposed FG-phobic surface[3]−which is very typical in a cellular context.

### Discussion

The permeability barrier of nuclear pore complexes (NPCs) can be described as a smart sieve made of cohesively interacting FG repeat domains. This sieve appears narrow-meshed for macromolecules

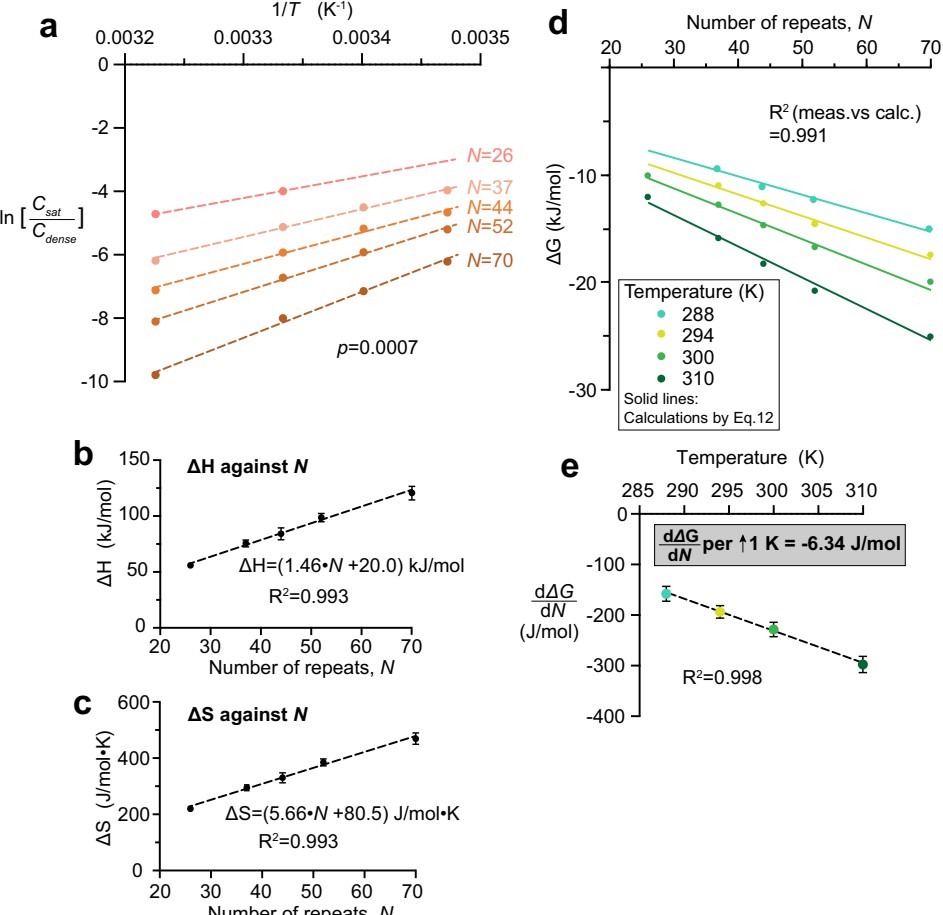

**Fig. 6 | Enthalpy, entropy and Gibbs free energy change in phase separation per FG repeat unit. a** ln ($C_{sat}/C_{dense}$) is plotted against $1/T$ for different numbers ($N$) of perfect repeats using dataset shown in Fig. 5. Each plot can be fitted with a linear function with a high R-squared value (>0.99). $\Delta H$ and $\Delta S$ for phase separation are derived from each linear fit (see Supp. Table 4). **b, c** $\Delta H$ and $\Delta S$ derived from best-fits in (**a**) were plotted against $N$, respectively. Data are presented as mean values with S.E. of fitting as error bars. **d** Dataset from Fig. 5d is converted to $\Delta G$ against $N$ by Eq. 2. All these data can be captured by a plot of Eq. 12 (solid lines). R-squared value here shows the degree of correlation between the dataset and calculations from Eq. 12. **e** Slopes of $\Delta G$ against $N$ (i.e., $\frac{d\Delta G}{dN}$) were derived from fitting the dataset in **d** and are plotted against temperature. Data are presented as mean values with S.E. of fitting as error bars.

without FG interactions but transiently widens for much larger objects that are bound to appropriate NTRs or engage themselves in facilitated translocation. A complementary perspective considers the barrier as a spatially localized good solvent for NTRs (selective phase), while inert FG-phobic macromolecules are poorly soluble in the phase. Understanding the biophysics of this phase is crucial for understanding transport selectivity and NPC function.

Several different FG domains are anchored to the NPC scaffold, but Nup98 FG domains appear particularly barrier-relevant. They occur in high copy numbers (48 per NPC[11,12]), are rather long (around 500 residues), remarkably depleted of charged residues, and poorly soluble in water. The latter manifests in a readily occurring phase separation even from dilute aqueous solutions. The underlying cohesive interactions are (mostly) hydrophobic with contributions of phenyl groups and aliphatic hydrocarbon moieties from the GLFG motifs and the inter-FG spacers[35].

The cohesion equilibrium cannot be described as a binary interaction with a simple dissociation constant because of the multivalency of the interaction originating from the multiplicity of cohesive repeat units. Moreover, a given cohesive unit likely interacts with several others. One consequence of such multi-level-multivalency is the high cooperativity of the interactions that leads to phase separation once a saturation concentration is reached/exceeded.

Previously, a similar description has been given for FG repeat films pre-immobilized on a plain surface[36–38]. We now provide the first quantitative thermodynamic description of cohesive interactions in a system that recapitulates nuclear transport selectivity faithfully[29]. It is based on a comprehensive, robust dataset that combines several orthogonal methods for deriving the relevant formulae and parameters (summarized in Tables 1 and 2). We started from a Nup98-derived FG domain comprising perfectly repeated cohesive units, which simplified the system considerably and allowed, for example, to quantitate cohesive interactions and provide $\Delta G$ values per interacting repeat. Remarkably, these values derived from orthogonal methods are very similar (200–250 J/mol at 294 K).

Computational modelling of IDPs and phase separation has recently become an important approach[76,86–93], including modelling of LCST-type phase separation[94,95], as well as FG domains[38,39,43]. Since all-atom explicit solvent simulations require extensive computational resources, coarse-grained modelling and analytical formulations are often more practical for studying phase separation involving long sequences and macromolecular assemblies. These approaches rely on parameterization, and thus we anticipate that our findings, including parameters and the pattern of change against environmental factors, will contribute to the application of these approaches. Notable previous examples are studies[76,91] on phase separation of the low-complexity domain from hnRNP A1 (A1-LCD), in which a stickers-and-spacers lattice model was parameterized by extensive experimental data, and the parameterized model revealed interesting quantitative

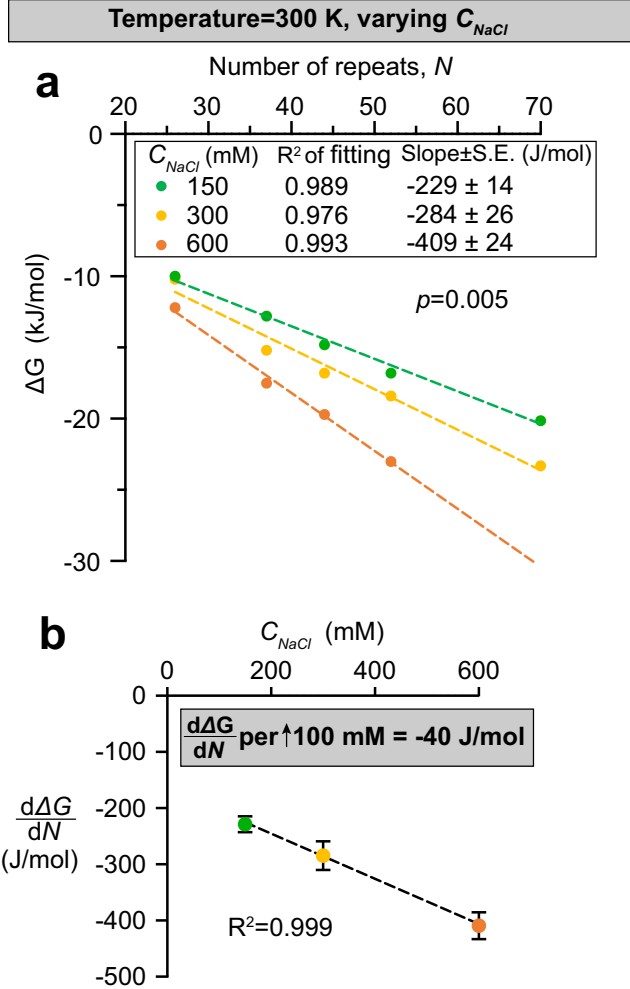

**Fig. 7 | Gibbs free energy change per FG repeat unit scales linearly with salt concentration. a** Datasets of *saturation concentrations* obtained at 150, 300 and 600 mM NaCl (shown in Fig. 5a, Supp. Fig. 4a and Supp. Fig. 4b, respectively) under the same temperature (300 K), are converted to $\Delta G$ against $N$ by Eq. 2. Each plot can be fitted with a linear function with a high R-squared value. **b** Slopes of $\Delta G$ against $N$ (i.e., $\frac{d\Delta G}{dN}$) derived from (**a**) are plotted against NaCl concentration ($C_{NaCl}$). Data are presented as mean values with S.E. of fitting as error bars.

insights. In addition, since we have shown that orthogonal methods revealed similar values of $\Delta G_{repeat}$, results from modelling of one context (e.g., partition experiment) will be likely transferrable to the modelling of others.

The quantification of cohesive interactions appears relevant also for a deeper understanding of transport selectivity. Mobile species that pass the phase need to transiently interrupt inter-FG cohesive interactions. If the needed $\Delta G$ cannot be compensated by favourable cargo-FG interactions, then exclusion results. Favourable interactions will result in an enrichment of the mobile species in the FG phase. As the $\Delta G$ is a continuum, a continuum of partition coefficients is also observed[3]. However, there are still several unknowns. First of all, we do not yet know the geometry of cohesive interactions, and how many of them need to get disengaged for a given mobile species to pass. It is plausible, however, that this number (and thus the energetic penalty) will scale with the mobile species' size and (FG-phobic) surface area.

An individual FG interaction is very weak, corresponding to 200–250 J/mol or about 0.1 $k_B \cdot T$, i.e., it can already be dissociated by thermal motion. This implies high local mobility[35], an incomplete occupancy of cohesive modules, and high availability of FG motifs for capturing NTRs and thus for NTR passage. However, avidity effects

through the 50 FG motifs in a Nup98 FG domain allow for a stable barrier against FG-phobic species. This may also explain why Nup98 FG domain homologues from evolutionarily very diverse species contain so many FG motifs[28]. An interesting parallel finding in other systems is that the pairwise interaction between "stickers" (Tyr and/or Phe residues), which drive the phase separation of Al-LCD, is of similar strength (0.3 $k_B \cdot T$)[91]. Although the phase separation of Al-LCD shows a UCST behaviour (i.e., with a negative value of $\Delta S$, indicating that entropy is against phase separation[76]) and Al-LCD has a different sequence composition than Nup98 FG domains, these numbers suggest that individual aromatic residues contribute weak affinities in phase separation.

It is well-documented that engineered elastin-like polymers (ELPs) phase separate with a linear relationship between the transition temperature and the logarithm of the ELP concentration[60,61]. This mirrors our observations for prf.GLFG$_{52\times12}$ and suggests that the two polymers assemble through similar driving forces.

There are fascinating parallels between the here described FG phase assembly and micelle formation from (non-ionic) detergent monomers, where also a (phase) transition occurs when the monomers reach the critical micelle concentration (*CMC*). Polyoxyethylene alkyl ethers[77–79,96–99] are the best studied non-ionic surfactants/detergents. They are often referred to as C$_i$E$_j$, where $i$ is the number of carbons in the alkyl chain (hydrophobic tail), and $j$ is the number of ethylene oxide units in the hydrophilic moiety (general formula C$_i$H$_{2i+1}$(OCH$_2$CH$_2$)$_j$OH).

The assembly of non-ionic micelles and of prf.GLFG$_{52\times12}$ FG phase are both driven by the hydrophobic effect. Both prf.GLFG$_{52\times12}$ and C$_i$E$_j$ are uncharged and thus their behaviours can be described by a simplified thermodynamic model (called "phase separation model" or "pseudo-phase separation model" in literature)[72–75]. The logarithm of the *CMC* of C$_i$E$_j$ detergents decreases linearly with the increase in the number of -CH$_2$- groups (=$i$) in the hydrophobic hydrocarbon tails[79,100–102]—just as the logarithm of $C_{sat}$ drops with the number of cohesive FG repeat units. This can be rationalized by the log-linear relationship between the equilibrium constant and $\Delta G$ that changes incrementally with the number of cohesive units -CH$_2$- groups and FG repeats, respectively.

The assembly of non-ionic micelles and the FG phase are both driven by entropy (positive values of $\Delta S$), and thus they exhibit LCST behaviours. A key characteristic is that $\Delta H$ and $\Delta S$ themselves are rather insensitive to the temperature (at least within the experimental ranges), such that linear relationships (between $1/T$ and ln*CMC* or ln$C_{sat}$, respectively) are observed in van't Hoff plots (Figs. 4a, 6a and refs. 77–79).

Moreover, it is well-documented that micellization of C$_i$E$_j$ is favoured by increasing salt concentration such that the Log of the CMC scales linearly with the salt concentration[103–105]. The same salt dependence was also observed for FG phase assembly (e.g., Fig. 3e).

Another similarity is that both FG phases and at least some micelles stained bright with environmentally sensitive Hoechst dyes[29,106]. This may suggest that both are hydrophobic structures and highly effective in shielding the dye against quenching by water molecules. From the practical aspect, such phenomenon may be applied for high-throughput measurements of saturation concentrations.

The extensive literature on micellization from the past decades also includes numerous theoretical modellings of the phase behaviour of C$_i$E$_j$[96,107–111], which may be inspiring for modelling the FG phase system.

Detergent micelles usually have a rather uniform size (in the range of nanometres) that is determined by the packing of the hydrophobic parts and by the geometry of the hydrophilic head groups that somehow "cap" the structure towards the water. However, phase separation of FG domains in free solution occurs without such a capping and thus without a restriction in size. This allows μm-sized FG particles to assemble by random seeding and subsequent growth, with surface tension leading to their roundish shape.

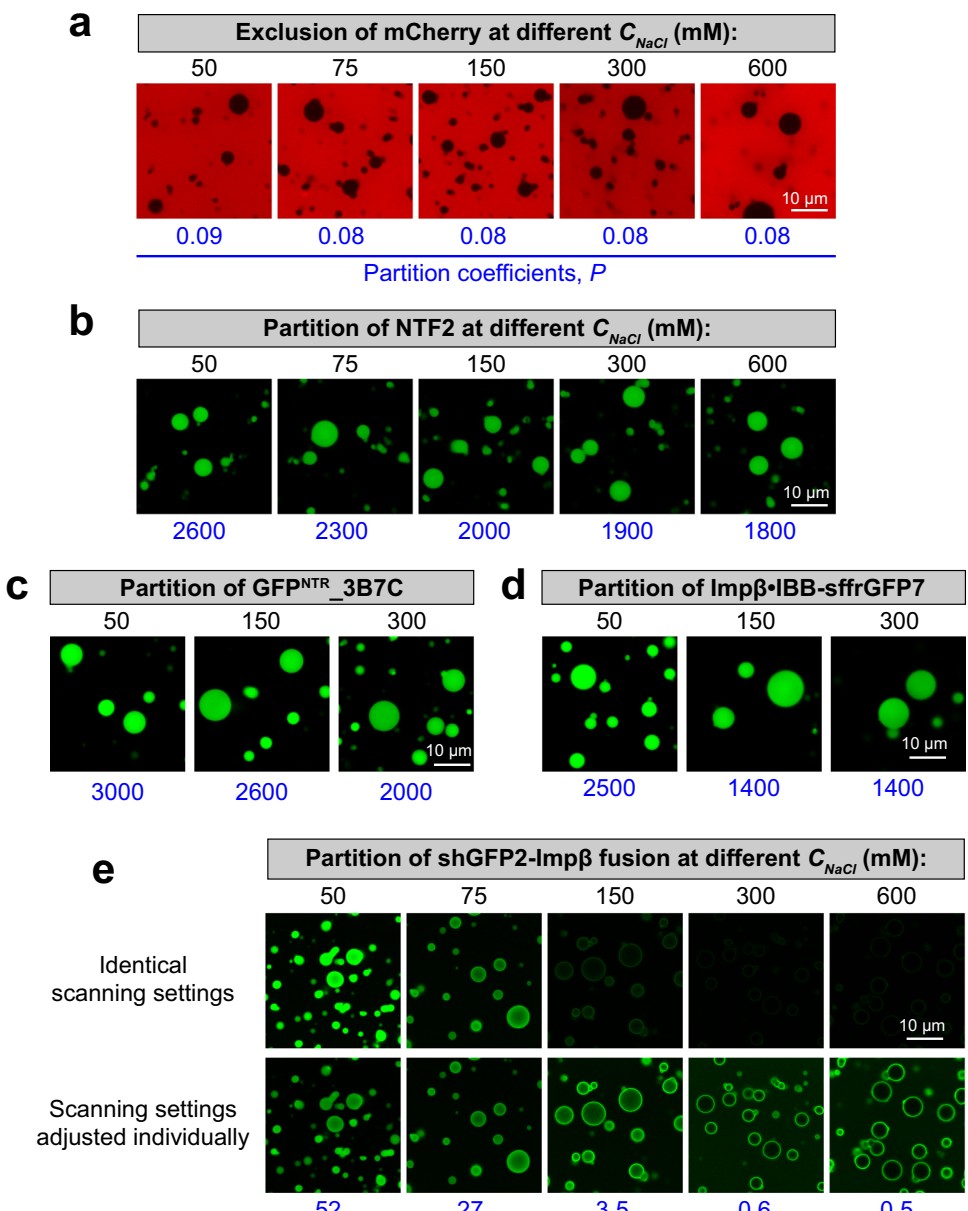

**Fig. 8 | Permeation selectivity of FG phase at different strength of FG-FG cohesion.** Partition coefficients of different permeation probes into in the FG phase formed by prf.GLFG$_{52\times12}$[+GLEBS] were measured at different indicated NaCl concentrations ($C_{NaCl}$), which modulate the strength of inter-FG cohesive interactions. These permeation probes include: an inert fluorescent protein mCherry of 27 kDa (**a**), NTF2 of 29 kDa (**b**), which is covalently coupled with Alexa488 for tracking, a tetrameric 110 kDa GFP$^{NTR}$_3B7C variant (**c**), a 130 kDa complex of Impβ and an FG-philic GFP variant sffrGFP7 containing an Impβ-binding (IBB) domain (**d**), and a shGFP2-Impβ fusion of 122 kDa (**e**). Identical scanning settings and image processing were applied for the same permeation probe. One more image set is included in **e**, whereas the scanning settings were adjusted between the set to capture the wide range of signals. Experiments in **a**–**e** were repeated two times independently with similar results, and the representative images are shown. Note that the binding of Impβ to IBB is sensitive to high salt and the Impβ-shGFP2 fusion allowed the assay in high salt condition (e.g., 600 mM).

NPCs contain roughly 12 distinct FG domains (of multiple copies) that are anchored to specific sites. Following the similarity to micelles, it is remarkable that the most cohesive ones (e.g., Nup98) appear rather central in location[12]. In contrast, less cohesive, charged ones are located at the nuclear (e.g., Nup153 or Nup1) or cytoplasmic sides (e.g., Nup159) as if they were capping the rather hydrophobic, super-cohesive core. This hydrophilic capping would then again be analogous to the hydrophilic head groups in a detergent micelle or a lipid bilayer. The resulting layered structure appears not only optimized from a thermodynamic perspective, but also the outer, more hydrophilic, and less densely packed layers might allow for a faster NTR capture and passivate the

barrier against certain non-specifically interacting, hydrophobic macromolecules.

## Methods
### Nomenclature
All perfectly repetitive sequences are named prf.GLFG$_{N\times12}$, where $N$ is the number of the repeat unit and 12 is the number of amino acid residues per repeat.

### DNA sequences of FG domain variants
DNA fragments encoding the prf.GLFG$_{N\times12}$ variants in a codon-optimized form were synthesized by *GenScript* and cloned into a

**Table 1 | Formulae and parameters derived in this study**

Formula Set 1 (from a DLS dataset):
$$\Delta G = a \cdot C_{NaCl} + b - T(c \cdot C_{NaCl} + d)$$
$$\ln\left(\frac{C_{sat}}{C_{dense}}\right) = \frac{a \cdot C_{NaCl} + b}{RT} - \frac{c \cdot C_{NaCl} + d}{R}$$

| Definitions | Parameters (best-fit values ± S.E.) and sub-formulae | Condition(s) |
|---|---|---|
| $\Delta G$: Gibbs free energy change (J/mol) for phase separation of prf.GLFG$_{52 \times 12}$<br><br>$C_{NaCl}$: concentration of NaCl in solvent (unit in mol/dm$^3$)<br>$a$: Enthalpy change per unit NaCl in solvent<br>$b$: Enthalpy when [NaCl] = 0<br>$c$: Entropy change per unit NaCl in solvent<br>$d$: Entropy when [NaCl] = 0<br>$T$: phase transition temperature (Kelvin)<br>$C_{sat}$: Saturation concentration (mol/dm$^3$)<br>$R$: gas constant (J/mol·K) | $a = 187{,}000 \pm 22{,}000\ \frac{\text{J} \cdot \text{dm}^3}{\text{mol}^2}$<br>$b = 102{,}000 \pm 7000\ \frac{\text{J}}{\text{mol}}$<br>$c = 688 \pm 81\ \frac{\text{J} \cdot \text{dm}^3}{\text{mol}^2 \cdot \text{K}}$<br>$d = 390 \pm 28\ \frac{\text{J}}{\text{mol} \cdot \text{K}}$<br><br>$\Delta\Delta G$ (J/mol) for phase separation of prf.GLFG$_{52 \times 12}$ per 1 K increase in temperature:<br>$\Delta\Delta G = -(c \cdot C_{NaCl} + d)$<br><br>$\Delta\Delta G$ (J/mol) for phase separation of prf.GLFG$_{52 \times 12}$ per unit (mol/dm$^3$) increase in ionic strength:<br>$\Delta\Delta G = a - T \cdot c$ | $N = 52$,<br>$310\,\text{K} \geq T \geq 275\,\text{K}^a$ |

Formula Set 2 (from a centrifugation dataset):
$$\Delta G = (w - T \cdot y) \cdot N - T \cdot z + x$$
$$\ln\left(\frac{C_{sat}}{C_{dense}}\right) = \frac{(w - T \cdot y) \cdot N - T \cdot z + x}{RT}$$

| Definitions | Parameters (best-fit values ± S.E.) and sub-formulae | Condition(s) |
|---|---|---|
| $\Delta G$: Gibbs free energy change (J/mol) for phase separation of a repetitive sequence with the number of repeats equal to $N$<br>$w$: Enthalpy change per repeat unit<br>$x$: Enthalpy constant<br>$y$: Entropy change per repeat unit<br>$z$: Entropy constant<br>$T$: phase transition temperature (Kelvin)<br>$C_{sat}$: Saturation concentration (mol/dm$^3$)<br>$C_{dense}$: FG phase concentration (mol/dm$^3$)<br>$R$: gas constant (J/mol·K) | $w = 1460 \pm 70\ \frac{\text{J}}{\text{mol}}$<br>$x = 20{,}000 \pm 3400\ \frac{\text{J}}{\text{mol}}$<br>$y = 5.66 \pm 0.27\ \frac{\text{J}}{\text{mol} \cdot \text{K}}$<br>$z = 80.5 \pm 13.1\ \frac{\text{J}}{\text{mol} \cdot \text{K}}$<br><br>$\Delta\Delta G$ (J/mol) for phase separation per 1 K increase in temperature<br>$\Delta\Delta G = -(y \cdot N + z)$<br><br>$\Delta\Delta G$ (J/mol) for phase separation per FG repeat increase<br>$\Delta\Delta G = w - T \cdot y$<br><br>$\Delta\Delta G$ (J/mol) for phase separation per FG repeat increase per 1 K increase in temperature<br>$\Delta\Delta G = -y$ | $C_{NaCl} = 0.15\,\text{mol/dm}^3$,<br>$70 \geq N \geq 26^a$,<br>$310\,\text{K} \geq T \geq 288\,\text{K}^a$ |

All numbers shown are rounded to three significant figures.
$^a$Validity ranges correspond to the ranges tested.

**Table 2 | Quantities obtained either by calculations based on formulae shown in Table 1 or by fitting the datasets specified**

| Quantity | Input values$^a$/conditions | Formulae/dataset | Output value |
|---|---|---|---|
| $\Delta\Delta G_{part.}$ per FG repeat increase | $C_{NaCl} = 0.15\,\text{mol/dm}^3$,<br>$T = 294\,\text{K}$ | Partition coefficient measurements (Fig. 1d, e) | −255 J/mol |
| $\Delta\Delta G_{part.}$ per FG repeat per 1 M increase in ionic strength | $T = 294\,\text{K}$ | | −282 J/mol |
| $\Delta G$ for phase separation | $N = 52$,<br>$C_{NaCl} = 0.15\,\text{mol/dm}^3$,<br>$T = 300\,\text{K}$ | Formula Set 1 | −17,600 J/mol |
| | | Formula Set 2 | −16,500 J/mol |
| Saturation concentration | | Formula Set 1 | $3.89 \times 10^{-6}\,\text{mol/dm}^3$ |
| | | Formula Set 2 | $6.12 \times 10^{-6}\,\text{mol/dm}^3$ |
| $\Delta\Delta G$ per 1 K increase in temperature | $N = 52$,<br>$C_{NaCl} = 0.15\,\text{mol/dm}^3$ | Formula Set 1 | −494 J/mol |
| | | Formula Set 2 | −375 J/mol |
| $\Delta\Delta G$ per 1 M increase in ionic strength | $N = 52$,<br>$T = 300\,\text{K}$ | Formula Set 1 | −19,300 J/mol |
| | | Centrifugation dataset (Fig. 3e) | −15,200 J/mol |
| $\Delta\Delta G$ per FG repeat increase | $C_{NaCl} = 0.15\,\text{mol/dm}^3$,<br>$T = 294\,\text{K}$ | Formula Set 2 | −202 J/mol |
| $\Delta\Delta G$ per FG repeat per 1 K increase in temperature | $C_{NaCl} = 0.15\,\text{mol/dm}^3$ | Formula Set 2 | −5.66 J/mol |
| $\Delta\Delta G$ per FG repeat per 1 M increase in ionic strength | $T = 300\,\text{K}$ | Centrifugation dataset (Fig. 7b) | −403 J/mol |

$^a$Input values for calculations are selected such that outputs from orthogonal formulae/datasets (DLS, centrifugation and partition datasets) can be compared. Note the high consistency of numbers derived from orthogonal methods.

bacterial expression vector for overexpression and purification (see below). See Supp. Table 5 for a complete list of the plasmids and the sources.

### Recombinant protein expression and purification

**prf.GLFG$_{52 \times 12}$.** *E. coli* NEB Express cells transformed with a plasmid, encoding the protein fused to a N-terminal histidine tag with a SUMO cleavage site (Supp. Table 5), were allowed to grow in a TB medium at 30 °C until OD$_{600}$ reached 3. Expression of the target protein was induced by 0.4 mM IPTG, 30 °C for 14 h. Then the protein was extracted from bacterial inclusion bodies with 4 M guanidine hydrochloride, 50 mM Tris/HCl pH 7.5, 10 mM DTT, purified by Ni-chromatography, cleaved by a SUMO protease (*Bd*SENP1[112]), re-solubilized with 30% acetonitrile and lyophilized, as previously described[35].

Lyophilized powder was weighed by a *Sartorius* ME235P analytical balance (S.D. was typically <10%) and stored at −20 °C until use.

**prf.GLFG$_{70×12}$, prf.GLFG$_{44×12}$, prf.GLFG$_{37×12}$.** Procedures were the same as that for prf.GLFG$_{52×12}$.

**prf.GLFG$_{26×12}$, prf.GLFG$_{18×12}$, prf.GLFG$_{13×12}$.** Expression was the same as above. However, unlike prf.GLFG$_{52×12}$, the target proteins did not form inclusion bodies in the bacterial host and remained in the soluble fraction after induction. For each, the soluble fraction was isolated by ultracentrifugation of the cell lysate (buffered in 50 mM Tris/HCl pH 8.0, 300 mM NaCl, 20 mM imidazole) and then was applied to a Ni(II) chelate column. The column was washed extensively in 50 mM Tris/HCl pH 8.0, 300 mM NaCl, 20 mM imidazole, 20 mM DTT and then in protease buffer: 50 mM Tris/HCl pH 7.5, 150 mM NaCl, 5 mM DTT. 50 nM bdSENP1 in protease buffer was applied for overnight on-column cleavage[113]. The cleaved target protein was eluted from the column with protease buffer, re-buffered to 30% acetonitrile by a PD10 Sephadex column (GE Healthcare), and lyophilized.

**prf.GLFG$_{7×12}$.** Procedures were mostly the same as that for prf.GLFG$_{26×12}$, except that the cleaved target protein from the Ni-column was re-buffered by a PD10 Sephadex column to the buffer conditions in the assays.

**prf.GLFG$_{52×12}$[+GLEBS].** The protein domain was recombinantly expressed as a histidine-tagged form in bacteria and purified as described before[29].

**Wild-type Nup98 FG domains.** The protein domains were recombinantly expressed as a histidine-tagged form in bacteria and purified as described before[28].

**NTRs, GFP variants and mCherry.** Most were expressed as His-tagged-fusions (Supp. Table 5) and purified by native Ni(II) chelate chromatography, as described previously[3,28]. Elution was performed by on-column SUMO protease cleavage[112].

**Fluorescent-labelling**
Each of the FG repeats for labelling (listed in Supp. Table 5) was expressed with an additional C-terminal Cysteine residue. For each, the lyophilized, purified protein was dissolved in 2 M guanidine hydrochloride and the protein was allowed to react with Atto488-maleimide (ATTO-TEC, Germany) at 1:1 molar ratio at pH 6.8. The labelled protein was further purified by gel filtration on a PD10 Sephadex column and quantified by the absorbance of Atto488.

*Quenched Atto488-maleimide in* Fig. 1: Atto488-maleimide was incubated with a 10× excess molar ratio of free L-cysteine for 1 h for quenching before the measurement.

**Measurement of partition coefficients and confocal laser scanning microscopy**
**Partition of FG repeats.** A solution of a given Atto488-labelled prf.GLFG$_{N×12}$ (as "guest") was prepared by diluting the stock in pre-cooled assay buffer (20 mM sodium phosphate (NaPi) pH 6.8, 150 mM NaCl, 5 mM DTT or [NaCl] specified in the figures) on ice. Concentrations were 3, 3, 2.5, 1, 1, 0.5, 0.5, 0.2 and 0.125 μM for $N = 0$ (only quenched Atto488-maleimide), 7, 13, 18, 26, 37, 44, 52 and 70, respectively. Note that the concentration of each guest was kept below the saturation concentration of the guest, such that the partition coefficient is independent of the guest concentration. At this point phase separation was suppressed by the low temperature on ice. Then 1 μl of a stock of the "host", unlabelled prf.GLFG$_{52×12}$[+GLEBS][29] (1 mM or 66 μg/μl protein in 4 M guanidine hydrochloride) was rapidly diluted with 200 μl of the solution containing the guest. 30 μl of the

mixture was placed on a collagen-coated micro-slide 18-well (IBIDI, Germany) and incubated at 21 °C to allow phase separation. Atto488 signal was acquired with 488 nm excitation with a Leica SP5/SP8 confocal scanning microscope (software: Leica Application Suite X 3.3.0.) equipped with a × oil immersion objective and hybrid detectors (standard mode, in which nonlinear response of the detector was auto-corrected) at 21 °C. Scanning settings were adjusted individually to cover wide dynamic ranges. Partition coefficient of the guest into the host (taken as signal inside FG phase: signal in buffer) was quantified as previously described[29]. prf.GLFG$_{52×12}$[+GLEBS] was chosen as the host because of its low saturation concentration (0.3 μM at below 600 mM NaCl, Supp. Fig. 1), so that the partition of the guest may not be interfered with the host in the aqueous phase. In Fig. 1b, brightness of the images was adjusted individually to normalize the brightness of the FG phase for comparison.

**Partition of NTRs/mCherry/GFPs.** As described previously[29]: For each measurement 2 μl of a 1 mM FG domain stock solution was rapidly diluted with 100 μl "NTR-permeation assay buffer" (50 mM Tris/HCl pH 7.5, 5 mM DTT and 50–600 mM NaCl as specified in the figures) to induce phase separation and 7.5 μl of the suspension was mixed with 22.5 μl substrate containing 1 μM NTR or 1 μM GFP or 5 μM mCherry in the same NTR-permeation assay buffer. The resulting mixture ([FG domain] = 5 μM) was placed on a collagen-coated micro-slide 18-well. FG particles were allowed to sediment for 60 min to the bottom of the slide at 21 °C and then imaged as described above.

**Detection of temperature-dependent phase transition by dynamic light scattering (DLS)**
In general, lyophilized powder of prf.GLFG$_{52×12}$ was dissolved to a concentration of 1 mM (equivalent to 58 μg/μl protein) in 2 M guanidine hydrochloride (called a stock solution). For each measurement, 1 μl of a freshly prepared stock solution was rapidly diluted with filtered NaCl solutions containing 20 mM NaPi pH 6.8, to protein concentrations and NaCl concentrations stated in the figures. For assay protein concentrations >50 μM, lyophilized powder of prf.GLFG$_{N×12}$ was dissolved in the NaCl solutions on ice (if necessary, a mild sonification was applied) and serially diluted. 10 μl of each solution was analysed in a closed cuvette using a DynaPro NanoStar DLS instrument (Wyatt Technologies). To acquire temperature-dependent phase separation curves, the temperature was automatically raised by 1 °C per min, typically from 2 to 40 °C. DLS signal was acquired continuously. Phase transition was indicated by a sharp increase in the light scattering intensity (typically from a threshold to at least a triple increase of the signal, which indicates clear deviation from monomeric states), and the transition temperature was rounded up to the nearest 0.1 °C. The Dynamics 7.1.5 software was used for autocorrelation analysis. This assay can detect temperature-dependent phase transition of prf.GLFG$_{52×12}$ at a concentration as low as about 2 μM. Three independent datasets for each condition were averaged, and standard deviations (S.D.) were shown as error bars in Fig. 3d.

**Analysis of phase separation by centrifugation (centrifugation assay)**
In general, for each, 1 μl of a fresh stock of prf.GLFG$_{N×12}$ (1-4 mM protein in 2 M guanidine hydrochloride) was rapidly diluted with assay buffer (typically 20 mM NaPi buffer pH 6.8, 5 mM DTT, 150 mM NaCl or [NaCl] specified in the figures), to the concentration stated in the figures. For assay protein concentrations >50 μM, lyophilized powder of prf.GLFG$_{N×12}$ was dissolved in the NaCl solutions on ice (if necessary, a mild sonification was applied) and serially diluted. After incubation at the temperature specified in the figures for 1 min, the FG phase (insoluble content) was pelleted by centrifugation (21,130 × *g*, 30 min, using a temperature-controlled Eppendorf 5424R centrifuge equipped with a FA-45-24-11 rotor) at the specified temperature. Equivalent ratio

of the pellet (condensed FG phase) and supernatant was analysed by SDS-PAGE/Coomassie blue-staining (the exact amount loaded for SDS-PAGE was adjusted individually such that the loaded amount of pellet + supernatant = 4.1 µg, unless specified). Saturation concentration of a given sample was taken as the concentration that remained in the supernatant, which was estimated with a concentration series loaded onto the same gel. This assay can quantify saturation concentrations as low as about 0.1 µM. All experiments were repeated independently at least two times with similar results, and the representative gel images are shown. Full scans of gels with molecular weight markers are provided in the Source data file.

### Data fitting
Fittings were by the least squares method by Microsoft Excel 16.42. Means of each dataset of replicates were fitted. R-squared values were calculated by the same software. $p$-values: two-tailed $p$-values were computed by Analysis of Covariance (ANCOVA) (in GraphPad Prism 9.2.0) to assess for each dataset if the differences in slopes of the linear fits are statistically significant (testing the null hypothesis that the slopes are all identical). No adjustment was made for multiple comparisons.

### Reporting summary
Further information on research design is available in the Nature Research Reporting Summary linked to this article.

## Data availability
The data that support the findings of this study are provided in the Supplementary Information or Source data file. Source data are provided with this paper.

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

## Acknowledgements
The authors thank Christian Griesinger and Eszter Najbauer for insightful discussions, as well as the Max-Planck-Gesellschaft and the Deutsche Forschungsgemeinschaft (SFB 860 and SFB 1190 to D.G.) for funding.

## Author contributions

S.C.N. planned and conducted all the experiments. D.G. conceived the overall concepts of the study. All authors contributed to experiment design, data analysis, interpretation, and manuscript writing.

## Funding

## Competing interests

The authors declare no competing interests.
