## [Peer Review File · Nature Communications]

REVIEWER COMMENTS

Reviewer #1 (Remarks to the Author):

The authors show that Nup98 FG domains present an LCST-type phase separation behaviour and provide a quantitative description of this behaviour by a comprehensive experimental dataset and a thermodynamic model derived from theories of micellization. The methodology performed is sound, and the work supports the conclusions. However, the manuscript requires a major revision before it can be accepted for publication. The comments for revision/clarifications are detailed below.

Major Comments:

1. The authors have stated that the engineered Nup98 FG domain, called prf.GLFG52x1223, captures the properties of the original FG domain very well. Since the entire work of the manuscript depends on this, it is essential that data (or reference) be provided to corroborate the claim.
2. To study phase separation, a pH of 6.8 was maintained. Why so? Similarly, 20 μ M of prf.GLFG52x1223 was taken (line 171, 271). Why is that?
3. What could be the reason for the changes experienced by the phase transition due to variation in temperature and salt concentration?
4. Line 358 states that “partition of artificially optimized FG-philic species are not significantly affected by the salt concentration.” This is followed by line 362, stating that “optimized FG-philic interactions strengthen with increasing salt concentration.” This seems a bit confusing and requires clarity.
5. According to the authors, why are there so many similarities between FG phase assembly and micellization of non-ionic surfactants?
6. What could be the consequences of the findings of this paper, like $\ln CCFG \propto -CNaCl$, for example, in vivo?

Minor Comments:

1. The abbreviated forms of certain words should be expanded, like NTF2.
2. Figure 2/Figure 4 may be rearranged to make it easier to follow from one figure to the next.

Reviewer #2 (Remarks to the Author):

This paper presents very interesting and useful data on the phase equilibrium of Nup98 FG domains. While the data appear very reliable, their theoretical interpretation -- which constitutes the major contribution of this work -- is problematic. The authors will need to reanalyze their data in a rigorous theoretical framework, rather than in a heuristic way as they do now. My Major comments are as follows.

1. Equation (6), which is the starting point of their data analysis is questionable. What is the meaning of this DG? That is, this DG is the free energy difference between what and what, or free energy change from which state to which state? The important initial and final states for defining this DG are never specified. More importantly, this equation is physically wrong. Here is the correct physics. The chemical potential, in either the condensed or the dilute phase, can be generally written as:

$$\mu = \mu_{id} + \mu_{ex}$$

where the first term represents the ideal part and the second term represents the excess part. The ideal part can be expressed as

$$\mu_{id} = k_B T \ln(C/C_0)$$

where C_0 is an unimportant constant. In the dilute phase, we can neglect μ_{ex} , so

$$\mu_{dil} = k_B T \ln(C_{dil}/C_0)$$

where C_{dil} represents the concentration in the dilute phase, which is known as the saturation concentration (or threshold concentration; see next comment). In the condensed phase, we have

$$\mu_{con} = k_B T \ln(C_{con}/C_0) + \mu_{ex}$$

where C_{con} represents the concentration in the condensed phase, and μ_{ex} now represents the excess chemical potential in the condensed phase.

At phase equilibrium, the chemical potentials in the two phases are equal, leading to

$$k_B T \ln(C_{\text{dil}}/C_{\text{con}}) = \mu_{\text{ex}}$$

This is the corrected equation. In essence, their Equation (6) is missing C_{con} , the concentration in the condensed phase. In the correct analysis, the authors need to put the concentration in the condensed phase into the above equation. The μ_{ex} is equivalent to their dG , and represents the excess chemical potential due to the non-ideal behavior of the condensed phase. An introduction to the foregoing theoretical analysis can be found in the following paper:

PMID: 29716768.

2. Another issue that needs to be corrected is the term "critical concentration" -- in the phase transition literature, this term has a very specific meaning, i.e., it is the concentration at the critical point. The corresponding temperature is called the critical temperature. The authors have used the term "critical concentration" to denote the concentration of the dilute phase, away from the critical point. The correct terminology is "saturation concentration", and sometimes also called the threshold concentration (e.g., PMID: 31506351). Other papers have misused the term "critical temperature", but we need to stop such misuse.

3. Continuing the above point, in p. 3, the authors correctly stated, notwithstanding the misused terminology, that the threshold concentration is an inverse measure of the phase separation propensity. The validity of this statement has been established in a computational study of UCST systems in PMID: 29712961 and this measure is used in PMID: 31506351.

4. The explanation presented here for the passage of NTRs is very similar to the mechanism of strong-attraction regulators in promoting phase separation (PMID: 31506351) -- these macromolecular regulators partition into the condensed phase and replace driver-driver attraction with stronger driver-regulator attraction.

5. In p. 5, the authors correctly noted the importance of

"cohesive interactions", but stated that they are too weak to be measured as a binary interaction. In fact, such weak interactions can be quantified by the second virial coefficient, which essentially is a measure of binary interactions (see, e.g., PMID: 31490691). The relative strengths of FG self-interaction and FG-NTR cross-interaction can be quantified by comparing the second virial coefficient and the cross second virial coefficient (PMID: 34241191).

6. Lastly, a technical point: in Fig. 1b,c, illustrative DLS data are shown. How did the authors determine a precise transition temperature from such data? E.g., did they fit the DLS data to a model for thermal transition and obtain the transition temperature from such a fit, or did they identify the first "big" jump in scattering intensity at the transition temperature -- in that case, how big is "big"?

Reviewer #3 (Remarks to the Author):

This manuscript by Ng and Goerlich is a beautiful, quantitative piece of work that characterizes and quantifies the phase behavior of a synthetic FG domain that is derived from the major cohesive FG domain of the nuclear pore complex, i.e., the Nup98 FG domain. The authors conceive of their experiments and analyze their data in a thermodynamic framework borrowed from surfactant micellation. The use of a synthetic FG domain made up of perfect repeats has several advantages; by altering the number of repeats, the authors are able to extract important parameters, including the interaction strength of a single FG repeat. They further quantify partitioning of shorter versions of their repeat polymer into the dense phase of the long version. Thereby, they gain insight into the thermodynamics of partitioning, an important parameter for transport receptor-facilitated transport through the NPC, which requires transient breaking of cohesive FG-FG interactions. They extend their study to natural FG domains from many different species and show that their results hold.

The manuscript is written beautifully without hyperbole and can be followed easily. What is missing are acknowledgements of previous work on similar systems.

1. Elastin-like peptides have been studied in detail for years before phase separation became recognized as a physiologically relevant assembly mechanism in cells. The types of measurements and analyses presented in Figure 1b-e, Figure 2, Figure 4 and 5 have been pioneered previously for elastin-like peptides by the labs of John Correia and Ashutosh Chilkoti. E.g. the linear relationship of the transition temperature vs. the logarithm of the biopolymer concentration is well known from the elastin field (Lyons et al. *Biophys J* 104: 2009-2021, 2013).

Perhaps Ng and Goerlich have reinvented these methods from scratch. Whether they were aware of them or not, this literature needs to be cited and the results presented here put into context. This doesn't diminish the careful work done in this current manuscript

2. The authors determine an "individual FG interaction strength" from their thermodynamic analysis. It is of note that this has also recently been accomplished for a UCST system, where the sticker-sticker interaction strength has been determined (Martin et al. *Science* 367: 694-699, 2020; Bremer et al. *bioRxiv* 2021 <https://www.biorxiv.org/content/10.1101/2021.01.01.425046v1>). In fact, Ng and Goerlich say in the discussion that these parameters could be used in coarse-grained simulations, which is exactly

what has been done in the hnRNPA1 system in the indicated papers (which are not cited). The weak affinities extracted in the current work and in the papers above are in good agreement.

3. Measurements of the partition coefficients of variants with different repeat number into Nup98 condensates is an interesting experiment. What is missing is the characterization of linkage effects, i.e., by how much is phase separation stabilized or destabilized when increasing concentrations of the short FG repeats titrated in? In fact, what would really move the field's understanding of the role of phase separation in the NPC forward would be a test of the selective phase model by the measurement of linkage effects from binding of NTRs to FG-Nups. If the FG-Nup phase is a good solvent for NTRs and they therefore partition into the Nup dense phase, this should result in a measurable effect on the driving force for phase separation. I think the authors have quantified the necessary parameters to be able to tell whether the measured stabilization of the FG-Nup condensates in the presence of NTRs would be in agreement with the selective phase model.

Minor comments:

1. The authors strategically use a perfect repeat protein for their study. That does not make this a homopolymer and should not be referred to as such. The reason that homopolymer models are able to describe the phase behavior of such systems has been shown to be caused by the uniform distribution of sticker residues in the sequence (Martin et al. *Science* 367: 694-699, 2020).

2. The authors are pointing to examples of proteins undergoing LCST phase separation, specifically IDPs. They are pointing to Pab1, but presumably the driving force for phase separation is encoded in its folded domains, with the IDR merely modulating the phase behavior.

3. What is ice temperature?

4. P. 10: Given that the sequences have LCST behavior, it is not "remarkable" that phase separation for all variants was disfavored by lowering the temperature. It is expected.

5. The authors propose that avidity effects from the ~50 FG motifs per chain allow for a stable barrier against FG-phobic species. How many FG motifs in any given chain are engaged in interactions at any given time and are the resulting back-on-the-envelope calculations in agreement with the observed cohesion?

Major changes to the manuscript

- We have modified/ simplified our equation for the free energy change during phase separation (now Equation 2). It now explicitly considers the protein concentration in the dense phase, as suggested by Reviewer #2.
- We have changed the order of the results and now show the partitioning of FG domains into host FG phases first. Since these experiments directly measure concentration ratios between dense and dilute phases, this is a more intuitive entry into the results.
- We have expanded the introduction and the results to emphasize the findings of our previous work, which validated the functionality of the perfectly repetitive Nup98 FG domain. This addresses the concern raised by Reviewer #1.
- We have expanded a paragraph in the results that details the difficulties in measuring the cohesive interactions directly and the necessity of our approach. This addresses a comment from Reviewer #2.
- We have expanded the discussion to include further references suggested by Reviewer #3.

Answers to the reviewers' comments (for clarity, the authors repeat reviewers' points in blue in front of each reply)

Reviewer #1 (Remarks to the Author): The authors show that Nup98 FG domains present an LCST-type phase separation behaviour and provide a quantitative description of this behaviour by a comprehensive experimental dataset and a thermodynamic model derived from theories of micellization. The methodology performed is sound, and the work supports the conclusions. However, the manuscript requires a major revision before it can be accepted for publication. The comments for revision/clarifications are detailed below.

Major Comments:

1. The authors have stated that the engineered Nup98 FG domain, called prf.GLFG_{52x1223}, captures the properties of the original FG domain very well. Since the entire work of the manuscript depends on this, it is essential that data (or reference) be provided to corroborate the claim.

We have indeed extensively characterized prf.GLFG_{52x12} in an earlier paper (Ng et al., 2021), demonstrating that prf.GLFG_{52x12} captures the essential functional (barrier-like) properties of the original MacNup98A FG domain and NPCs very well. We also found that prf.GLFG_{52x12} and MacNup98A FG phases show very similar local dynamics in solution and solid-state NMR (Najbauer et al., 2022). We adjusted the text to make the references to these published data clearer.

2. To study phase separation, a pH of 6.8 was maintained. Why so?

This near physiological pH was chosen to match the pH and buffer ion (phosphate) of earlier NMR datasets (Najbauer et al., 2022). Given the uncharged nature of the sequence, pH variations (around neutral) have actually little effect.

Similarly, 20 μM of prf.GLFG_{52x1223} was taken (line 171, 271). Why is that?

We estimated saturation concentrations by a centrifugation method. This required phase separation to happen and thus the experimental FG domain concentration had to exceed the saturation concentration. 20 μM of prf.GLFG_{52x12} was used in this experiment because it ensured that phase separation at all the NaCl concentrations listed in Fig. 3e (of the current version) had happened (as suggested by the dataset shown in Fig. 3d).

3. What could be the reason for the changes experienced by the phase transition due to variation in temperature and salt concentration?

Inter FG-cohesion (and thus FG phase separation) are driven by hydrophobic interactions, and these get stronger with higher temperature and higher NaCl concentrations. This relates to entropic costs for re-organizing water molecules (with dissolved salt) around hydrophobic moieties. Less water needs to be re-organized when hydrophobic groups interact with each other and shield each other against the solvent. The entropic costs (ΔS) scale with temperature (T), as described by: $\Delta G = \Delta H - T\Delta S$ (ΔG , free energy change; ΔH = enthalpy change). We have modified a paragraph in the Results to make this clearer.

4. Line 358 states that “partition of artificially optimized FG-philic species are not significantly affected by the salt concentration.” This is followed by line 362, stating that “optimized FG-philic interactions strengthen with increasing salt concentration.” This seems a bit confusing and requires clarity.

Agreed. The rationale is that hydrophobic interactions get stronger with higher salt and that this should apply not only to hydrophobic interactions between FG repeats but also to hydrophobic NTR·FG interactions. We have re-written the entire section for improved clarity.

5. According to the authors, why are there so many similarities between FG phase assembly and micellization of non-ionic surfactants?

Perhaps two aspects: both processes are driven by hydrophobic interactions and the interacting molecules (essentially) lack charges. For the non-ionic surfactants, the latter is trivial, but the extreme depletion of charges in Nup98 FG domains is something really very special among phase-separating IDPs.

6. What could be the consequences of the findings of this paper, like $\ln \text{CCFG} \propto -C\text{NaCl}$, for example, *in vivo*?

The focus of our study was to probe the properties of the NPC permeability barrier (respectively of its simplest version) and thus to gain biophysical insights into assembly- and transport mechanisms. Measuring the impacts of salt and temperature was part of this probing.

A broader question is how NPCs respond *in vivo* to changes in temperature and salt concentrations. Temperature and salt effects on phase separation: Since FG domains are anchored to NPCs at far higher concentrations than needed for phase separation (Schmidt and

Görlich, 2015), we assume that the cohesion-driven assembly step of the permeability barrier occurs robustly over a wide range of conditions.

Effects on the stringency of the permeability barrier: Cellular temperatures of homeotherm organisms (mammals, birds) will probably vary less than needed for noticeable effects. The same applies to intracellular salt concentrations, which are also tightly regulated. Increasing intracellular salt to say 300 mM would correspond to extreme stress with probably lethal outcome.

Sure enough, many organisms tolerate wide temperature changes. Plants (or insects), for example, can experience differences of up to 30°C between a cool, cloudless night and full sun on the next day. Keeping cellular infrastructure functional requires effective stress responses and temperature compensation mechanisms. We assume that this also applies to NPCs. One way to achieve robustness to temperature changes would be to “mix” hydrophobic and ionic interactions, and this might be one reason why several peripheral FG domains of NPCs contain charged residues. The loading of the FG phase with (charged) clients (NTRs) might also have temperature-compensating effects, in particular, if this client spectrum can be adapted by stress responses.

Minor Comments:

1. The abbreviated forms of certain words should be expanded, like NTF2.

The vast majority of literature considers NTF2 as a proper name. However, we have expanded the abbreviated forms of, e.g., NTF2 and IBB.

2. Figure 2/Figure 4 may be rearranged to make it easier to follow from one figure to the next.

We have rearranged the panels in Figure 2 (=Figure 4 of the current version).

Reviewer #2 (Remarks to the Author):

This paper presents very interesting and useful data on the phase equilibrium of Nup98 FG domains. While the data appear very reliable, their theoretical interpretation -- which constitutes the major contribution of this work -- is problematic. The authors will need to reanalyze their data in a rigorous theoretical framework, rather than in a heuristic way as they do now. My Major comments are as follows.

1. Equation (6), which is the starting point of their data analysis is questionable. What is the meaning of this DG? That is, this DG is the free energy difference between what and what, or free energy change from which state to which state? The important initial and final states for defining this DG are never specified. More importantly, this equation is physically wrong. Here is the correct physics. The chemical potential, in either the condensed or the dilute phase, can be generally written as:

$$\mu = \mu_{id} + \mu_{ex}$$

where the first term represents the ideal part and the second term represents the excess part. The ideal part can be expressed as

$$\mu_{id} = k_B T \ln(C/C_0)$$

where C_0 is an unimportant constant. In the dilute phase, we can neglect μ_{ex} , so

$$\mu_{dil} = k_B T \ln(C_{dil}/C_0)$$

where C_{dil} represents the concentration in the dilute phase, which is known as the saturation concentration (or threshold concentration; see next comment). In the condensed phase, we have

$$\mu_{\text{con}} = k_B \cdot T \ln(C_{\text{con}}/C_0) + \mu_{\text{ex}}$$

where C_{con} represents the concentration in the condensed phase, and μ_{ex} now represents the excess chemical potential in the condensed phase.

At phase equilibrium, the chemical potentials in the two phases are equal, leading to

$$k_B \cdot T \ln(C_{\text{dil}}/C_{\text{con}}) = \mu_{\text{ex}}$$

This is the corrected equation. In essence, their Equation (6) is missing C_{con} , the concentration in the condensed phase. In the correct analysis, the authors need to put the concentration in the condensed phase into the above equation. The μ_{ex} is equivalent to their dG , and represents the excess chemical potential due to the non-ideal behavior of the condensed phase. An introduction to the foregoing theoretical analysis can be found in the following paper:

PMID: 29716768.

Thank you very much for the suggestions. We actually followed this logic already when analysing the partitioning of the Atto488-labelled size series of FG domain fragments into the host FG phase. The partition coefficient P corresponds to $C_{\text{con}}/C_{\text{dil}}$. Our equation (now equation 1) is essentially identical to the last equation of the reviewer (with the substitutions $R=N_A \cdot k_B$, $\Delta G= N_A \cdot \mu$, $P= C_{\text{con}}/C_{\text{dil}}$, and with a correction term G^0 that considers that the fluorescent label is weakly FG-philic). We now changed the order of the results and show the partition experiments first, simply because it is indeed the more intuitive set of experiments.

ΔG for phase separation: We now consider the protein concentration in the dense phase (C_{dense}) explicitly (now equation 2). This simplifies the math. In fact, we had measured C_{dense} in the prf.GLFG_{52x12} phase before and found a mass concentration of 260 mg/ml (Najbauer et al., 2022).

This mass concentration is also consistent with the typical concentration of FG phases assembled from native Nup98 FG domains of evolutionarily distant species (Schmidt and Görlich, 2015). So far, we observed no strong dependence of C_{dense} on the temperature and salt concentration within the experiment range. For example, we assembled the FG phase in different $[\text{NaCl}]$, in which 1.1% of all the FG domain molecules contain an Atto488-fluorophore (Fig.R1). We found that the intra-FG phase fluorescence (reflecting C_{dense}) does not change significantly even up to 1200 mM NaCl, despite variations between FG particles. Therefore, for simplicity, we set 4.5 mM to be the reference concentration. This approximation approach was also adopted recently by Bremer et al., 2022.

2. Another issue that needs to be corrected is the term "critical concentration" -- in the phase transition literature, this term has a very specific meaning, i.e., it is the concentration at the critical point. The corresponding temperature is called the critical temperature. The authors have used the term "critical concentration" to denote the concentration of the dilute phase, away from the critical point. The correct terminology is "saturation concentration", and sometimes also called the threshold concentration (e.g., PMID: 31506351). Other papers have misused the term "critical temperature", but we need to stop such misuse.

We have changed the term.

3. Continuing the above point, in p. 3, the authors correctly stated, notwithstanding the misused terminology, that the threshold concentration is an inverse measure of the phase separation propensity. The validity of this statement has been established in a computational study of UCST systems in PMID: 29712961 and this measure is used in PMID: 31506351.

This is an interesting point. We (and many other authors before us, e.g., Wang et al., 2018) consider this inverse relationship as a pragmatic and experimentally useful *definition* and *measure* of the phase separation propensity (e.g., when two experimental conditions or two phase-separating proteins are compared) and not as an assumption/ hypothesis to be tested. We have added a few references to document that this definition is generally adopted in the field of biological condensates.

4. The explanation presented here for the passage of NTRs is very similar to the mechanism of strong-attraction regulators in promoting phase separation (PMID: 31506351) -- these macromolecular regulators partition into the condensed phase and replace driver-driver attraction with stronger driver-regulator attraction.

This is an interesting parallel. However, we wish to point out that we proposed this mechanism 18 years earlier (Ribbeck and Görlich, 2001) than Ghosh et al., 2019 (PMID: 31506351). The current version already includes 113 references, and we are therefore hesitant to extend the list much further.

5. In p. 5, the authors correctly noted the importance of "cohesive interactions", but stated that they are too weak to be measured as a binary interaction. In fact, such weak interactions can be quantified by the second virial coefficient, which essentially is a measure of binary interactions (see, e.g., PMID: 31490691). The relative strengths of FG self-interaction and FG-NTR cross-interaction can be quantified by comparing the second virial coefficient and the cross second virial coefficient (PMID: 34241191).

The computational tool(s) described in PMID: 31490691 (Qin and Zhou, 2019) and 34241191 (Ahn et al., 2021) all consider interactions between folded domains and require appropriate structures files as inputs. This is not applicable to the here studied cohesive interactions, which occur between intrinsically disordered domains, for which no structure files are available. We do not know the valency of cohesive interactions. The cohesive units might interact as pairs, trimers, tetramers, etc. The interacting state might be a single defined one (unlikely) or a mixture of many (probably).

Our initial description of the difficulties was perhaps a bit too focused on the interaction strength, and lacked the issues of fuzziness and lack of structural information. We have now expanded the paragraph as follows: *“A challenge in measuring this parameter is that cohesive interactions are rather complex and associated with several unknowns. First, many cohesive units within an FG domain need to be considered. Second, given the repeats’ intrinsic disorder, cohesive interactions are probably fuzzy and heterogeneous. Finally, the valency with which elementary cohesive units interact is unknown; and possibly, this valency is not even fixed but flexible. Thus, standard biochemical dissociation constants appear to be a rather inadequate description.”*

6. Lastly, a technical point: in Fig. 1b,c, illustrative DLS data are shown. How did the authors determine a precise transition temperature from such data? E.g., did they fit the DLS data to a model for thermal transition and obtain the transition temperature from such a fit, or did they identify the first "big" jump in scattering intensity at the transition temperature -- in that case, how big is "big"?

We did not fit the DLS data to a model. We identified the first jump in scattering intensity as the transition temperature: from a threshold to an at least a threefold increase of the signal, which indicates clear deviation from monomeric states. Indeed, precise determination of transition temperature is limited by a number of technical factors, e.g., how fast did the temperature of the sample in the reaction chamber reaches the equilibrium and the recorded temperature, and how fast did the sample react to the temperature change. Errors of determination of transition temperature cannot be avoided. We were aware of that and have performed the complimentary experiment, in which we fixed the temperature and measured the saturation concentration at that temperature (as shown in Fig.3e of the current version). The consistency of both datasets suggests that our estimations of transition temperature are within a reasonable range. Moreover, in a log scale, on which we performed our calculations, such technical errors may not be significant. We have expanded the description of how we estimate the transition temperature in the Methods.

Reviewer #3 (Remarks to the Author):

This manuscript by Ng and Goerlich is a beautiful, quantitative piece of work that characterizes and quantifies the phase behavior of a synthetic FG domain that is derived from the major cohesive FG domain of the nuclear pore complex, i.e., the Nup98 FG domain. The authors conceive of their experiments and analyze their data in a thermodynamic framework borrowed from surfactant micellation. The use of a synthetic FG domain made up of perfect repeats has several advantages; by altering the number of repeats, the authors are able to extract important parameters, including the interaction strength of a single FG repeat. They further quantify partitioning of shorter versions of their repeat polymer into the dense phase of the long version. Thereby, they gain insight into the thermodynamics of partitioning, an important parameter for transport receptor-facilitated transport through the NPC, which requires transient breaking of cohesive FG-FG interactions. They extend their study to natural FG domains from many different species and show that their results hold. The manuscript is written beautifully without hyperbole and can be followed easily. What is missing are acknowledgements of previous work on similar systems.

Thank you for this enthusiastic summary. We apologize that we have missed the acknowledgements of some of the previous work. We have now improved the manuscript by referencing the suggested previous work of other systems (see below) to cover the wide perspectives of this current multidisciplinary work.

1. Elastin-like peptides have been studied in detail for years before phase separation became recognized as a physiologically relevant assembly mechanism in cells. The types of measurements and analyses presented in Figure 1b-e, Figure 2, Figure 4 and 5 have been pioneered previously for elastin-like peptides by the labs of John Correia and Ashutosh Chilkoti. E.g. the linear relationship of the transition temperature vs. the logarithm of the biopolymer concentration is well known from the elastin field (Lyons et al. *Biophys J* 104: 2009-2021, 2013).

Perhaps Ng and Goerlich have reinvented these methods from scratch. Whether they were aware of them or not, this literature needs to be cited and the results presented here put into context. This doesn't diminish the careful work done in this current manuscript

Already in the initial submission, we had cited three beautiful papers on elastin-like-peptides (ELPs), including two from the Chilkoti lab. We have now added Lyons et al., 2013 and extended the cross-referencing to five references on ELPs and three occurrences in the text.

Indeed, the linear relationship of the transition temperature vs. the logarithm of the biopolymer concentration is a striking similarity between the two systems (as reported by Meyer and Chilkoti, 2004 and in the present ms). This parallel is now explicitly cited. However, the data have been analyzed from different angles, as we interpreted the datasets from a perspective of free energy changes and integrated more parameters.

2. The authors determine an “individual FG interaction strength” from their thermodynamic analysis. It is of note that this has also recently been accomplished for a UCST system, where the sticker-sticker interaction strength has been determined (Martin et al. *Science* 367: 694-699, 2020; Bremer et al. *bioRxiv* 2021 <https://www.biorxiv.org/content/10.1101/2021.01.01.425046v1>). In fact, Ng and Goerlich say in the discussion that these parameters could be used in coarse-grained simulations, which is exactly what has been done in the hnRNPA1 system in the indicated papers (which are not cited). The weak affinities extracted in the current work and in the papers above are in good agreement.

Thanks for this reminder. The approach/ findings of these two studies indeed strongly support our arguments in the discussion. We also appreciated the fact that the weak affinities extracted

in the current work and in these studies are in good agreement. We have expanded the discussion accordingly.

3. Measurements of the partition coefficients of variants with different repeat number into Nup98 condensates is an interesting experiment. What is missing is the characterization of linkage effects, i.e., by how much is phase separation stabilized or destabilized when increasing concentrations of the short FG repeats titrated in? In fact, what would really move the field's understanding of the role of phase separation in the NPC forward would be a test of the selective phase model by the measurement of linkage effects from binding of NTRs to FG-Nups. If the FG-Nup phase is a good solvent for NTRs and they therefore partition into the Nup dense phase, this should result in a measurable effect on the driving force for phase separation. I think the authors have quantified the necessary parameters to be able to tell whether the measured stabilization of the FG-Nup condensates in the presence of NTRs would be in agreement with the selective phase model.

We performed the following two experiments as suggested. Firstly, we analyzed phase separation of our standard prf.GLFG_{52x12} when increasing concentrations of the shorter prf.GLFG_{26x12} were titrated in (Fig.R2). We found, however, that the phase separation (or saturation concentration) of prf.GLFG_{52x12} was not affected. This is perhaps not too surprising as the more multivalent species should dominate the phase separation process, and the shorter one should remain more soluble.

Fig.R2: 20 μM of prf.GLFG_{52x12} were allowed to phase separate in the presence of prf.GLFG_{26x12} of the indicated concentrations in 20mM NaPi pH6.8, 150mM NaCl and centrifuged at the same temperature (27°C / 300 K). SDS samples of the obtained pellets (FG phase), and supernatants (soluble content) were loaded for SDS-PAGE at equal ratio (7%), followed by Coomassie blue staining.

Secondly, we analyzed phase separation of prf.GLFG_{52x12} in the presence of different NTRs (Fig.R3). There was nothing striking. Importin β slightly increased the soluble fraction of prf.GLFG_{52x12}, perhaps by chaperoning the soluble FG domain. NTF2 and GFP^{NTR}3B7C had no effect, even though GFP^{NTR}3B7C partitioned so strongly into the FG phase that it must have become a structural component of the phase.

We would not exclude that NTRs might also favor phase separation if we used different FG domains, NTRs, NTR: FG domain ratios, salt or temperatures. However, exploring such parameter space is beyond the scope of the current study.

Fig.R3: 20 μ M of prf.GLFG_{52x12} were allowed to phase separate in the absence or presence of different NTRs in 20mM NaPi pH6.8, 150mM NaCl and centrifuged at the same temperature (27°C / 300 K). SDS samples of the obtained pellets (FG phase), and supernatants (soluble content) were loaded for SDS-PAGE at equal ratio (7%), followed by Coomassie blue staining.

Minor comments:

1. The authors strategically use a perfect repeat protein for their study. That does not make this a homopolymer and should not be referred to as such. The reason that homopolymer models are able to describe the phase behavior of such systems has been shown to be caused by the uniform distribution of sticker residues in the sequence (Martin et al. Science 367: 694-699, 2020).

Thanks for pointing out this. We have modified the descriptions.

2. The authors are pointing to examples of proteins undergoing LCST phase separation, specifically IDPs. They are pointing to Pab1, but presumably the driving force for phase separation is encoded in its folded domains, with the IDR merely modulating the phase behavior.

We have removed Pab1 as the examples.

3. What is ice temperature?

We refer to the low temperature ($\sim 0^\circ\text{C}$) when the samples were put on ice. We have modified the sentences to make it clearer.

4. P. 10: Given that the sequences have LCST behavior, it is not “remarkable” that phase separation for all variants was disfavored by lowering the temperature. It is expected.

We have modified the sentence.

5. The authors propose that avidity effects from the ~ 50 FG motifs per chain allow for a stable barrier against FG-phobic species. How many FG motifs in any given chain are engaged in interactions at any given time and are the resulting back-on-the-envelope calculations in agreement with the observed cohesion?

Well, the statement of a “stable barrier against FG-phobic species” is an observation (as seen from the perfect exclusion of mCherry from the FG phase); the reference to avidity effects is an attempt to reconcile this observation with the weakness of cohesive interactions, which corresponds to a fraction of $k\cdot T$ per repeat unit.

There are a few ways of estimating the occupancy of a single given repeat unit within the FG phase. First, we can use the dataset for partitioning fluorescently labelled FG domains of varying repeat number into a host phase. These data suggest that adding one repeat unit increases the partition coefficient by a factor of 1.11. The increase should correspond to the cohesive occupancy (CO) of that added repeat unit: $\text{CO}=(1.11-1)/1.11=0.1$.

The occupancy of cohesive interactions might be similar to that fraction. However, there are reasons to assume that this back-on-the-envelope calculation underestimates the occupancy and oversimplifies the system:

First, it is highly unlikely that cohesive interactions can be described as a single state. Instead, there are probably many ways how residues of one repeat can interact with residues of another repeat: transient hydrophobic contacts between aliphatic and/ or aromatic carbons, van der Waals interactions, or hydrogen-bondings. In fact, we directly observed such a variety of interactions by NMR (Najbauer et al., 2022; Figure 6). This comes with some difficulties in defining THE interacting state. Furthermore, the very high local concentration of 260 mg/ml already implies close spatial proximity between repeat units and suggests that most repeat units are somehow in contact with another one at any point in time. Indeed, we reported recently (Najbauer et al., 2022; Figure 4b) that the phenylalanines of the repeats are ~ 6 times (@ 24°C) less mobile in the FG phase than in bulk solvent. This suggests that most of them are engaged in interactions that increase rotational correlation times. Given the multi-state nature of the cohesive interactions, however, it is still well possible that a sizeable fraction of FG motifs remains available for NTR-binding. In the end, we will need atomistic simulations to answer the reviewer’s question.

Literature cited in this document:

- Ahn, S. H., Qin, S., Zhang, J. Z., McCammon, J. A., Zhang, J., & Zhou, H. X. (2021). Characterizing protein kinase A (PKA) subunits as macromolecular regulators of PKA RI α liquid-liquid phase separation. *J Chem Phys*, *154*(22), 221101.
- Bremer, A., Farag, M., Borchers, W. M., Peran, I., Martin, E. W., Pappu, R. V. et al. (2022). Deciphering how naturally occurring sequence features impact the phase behaviours of disordered prion-like domains. *Nat Chem*, *14*(2), 196-207.
- Ghosh, A., Mazarakos, K., & Zhou, H. X. (2019). Three archetypical classes of macromolecular regulators of protein liquid-liquid phase separation. *Proc Natl Acad Sci U S A*, *116*(39), 19474-19483.
- Lyons, D. F., Le, V., Bidwell, G. L., Kramer, W. H., Lewis, E. A., Raucher, D. et al. (2013). Structural and hydrodynamic analysis of a novel drug delivery vector: ELP[V5G3A2-150]. *Biophys J*, *104*(9), 2009-2021.
- Martin, E. W., Holehouse, A. S., Peran, I., Farag, M., Incicco, J. J., Bremer, A. et al. (2020). Valence and patterning of aromatic residues determine the phase behavior of prion-like domains. *Science*, *367*(6478), 694-699.
- Meyer, D. E., & Chilkoti, A. (2004). Quantification of the effects of chain length and concentration on the thermal behavior of elastin-like polypeptides. *Biomacromolecules*, *5*(3), 846-851.
- Najbauer, E. E., Ng, S. C., Griesinger, C., Görlich, D., & Andreas, L. B. (2022). Atomic resolution dynamics of cohesive interactions in phase-separated Nup98 FG domains. *Nat Commun*, *13*(1).
- Ng, S. C., Güttler, T., & Görlich, D. (2021). Recapitulation of selective nuclear import and export with a perfectly repeated 12mer GLFG peptide. *Nat Commun*, *12*(1), 4047.
- Qin, S., & Zhou, H. X. (2019). Calculation of Second Virial Coefficients of Atomistic Proteins Using Fast Fourier Transform. *J Phys Chem B*, *123*(39), 8203-8215.
- Ribbeck, K., & Görlich, D. (2001). Kinetic analysis of translocation through nuclear pore complexes. *EMBO J*, *20*(6), 1320-1330.
- Schmidt, H. B., & Görlich, D. (2015). Nup98 FG domains from diverse species spontaneously phase-separate into particles with nuclear pore-like permselectivity. *Elife*, *4*, e04251.
- Wang, J., Choi, J. M., Holehouse, A. S., Lee, H. O., Zhang, X., Jahnel, M. et al. (2018). A Molecular Grammar Governing the Driving Forces for Phase Separation of Prion-like RNA Binding Proteins. *Cell*, *174*(3), 688-699.e16.

REVIEWERS' COMMENTS

Reviewer #1 (Remarks to the Author):

I congratulate the authors for judiciously revising the manuscript. They managed to improve the manuscript and were able to reply to the queries adequately. I suggest accepting the manuscript now.

Reviewer #3 (Remarks to the Author):

The authors have addressed the comments satisfactorily and I recommend publication of this insightful manuscript.

Answers to the reviewers' comments (for clarity, the authors repeat reviewers' points in blue in front of each reply)

Reviewer #1 (Remarks to the Author):

I congratulate the authors for judiciously revising the manuscript. They managed to improve the manuscript and were able to reply to the queries adequately. I suggest accepting the manuscript now.

Thank you very much.

Reviewer #3 (Remarks to the Author):

The authors have addressed the comments satisfactorily and I recommend publication of this insightful manuscript.

Thank you very much.